# A novel decoy strategy for polymyxin resistance in *Acinetobacter baumannii*

**Jaeeun Park[1†], Misung Kim[1†], Bora Shin[1†], Mingyeong Kang[1], Jihye Yang[1], Tae Kwon Lee[2], Woojun Park[1]***

[1]Laboratory of Molecular Environmental Microbiology, Department of Environmental Science and Ecological Engineering, Korea University, Seoul, Republic of Korea; [2]Department of Environmental Engineering, Yonsei University, Wonju, Republic of Korea

**Abstract** Modification of the outer membrane charge by a polymyxin B (PMB)-induced PmrAB two-component system appears to be a dominant phenomenon in PMB-resistant *Acinetobacter baumannii*. PMB-resistant variants and many clinical isolates also appeared to produce outer membrane vesicles (OMVs). Genomic, transcriptomic, and proteomic analyses revealed that upregulation of the *pmr* operon and decreased membrane-linkage proteins (OmpA, OmpW, and BamE) are linked to overproduction of OMVs, which also promoted enhanced biofilm formation. The addition of OMVs from PMB-resistant variants into the cultures of PMB-susceptible *A. baumannii* and the clinical isolates protected these susceptible bacteria from PMB. Taxonomic profiling of in vitro human gut microbiomes under anaerobic conditions demonstrated that OMVs completely protected the microbial community against PMB treatment. A *Galleria mellonella*-infection model with PMB treatment showed that OMVs increased the mortality rate of larvae by protecting *A. baumannii* from PMB. Taken together, OMVs released from *A. baumannii* functioned as decoys against PMB.

*For correspondence:
wpark@korea.ac.kr

[†]These authors contributed equally to this work

Competing interests: The authors declare that no competing interests exist.

## Introduction

Pathogenic *Acinetobacter baumannii* strains cause urinary tract infections and ventilator-associated pneumonia (*Wright et al., 2014*). The incidence and prevalence of multidrug-resistant *A. baumannii* (MRAB), which are resistant to various antibiotics including sulbactam and tigecycline, threaten the effective prevention and treatment of MRAB-related infections (*Brauers et al., 2005*; *Perez et al., 2007*; *Alsan and Klompas, 2010*). The broad-spectrum β-lactam carbapenem antibiotics, such as imipenem, meropenem, and doripenem, are mainly used to treat severe MRAB infections (*Papp-Wallace et al., 2011*; *Kang et al., 2012*). Unfortunately, studies have reported the spread of MRAB clonal complex 92, which contains the *blaOXA-23* gene encoding a carbapenem-hydrolyzing class D β-lactamase (*Antunes et al., 2014*; *Yoon et al., 2017*). Mobile genetic elements, particularly *ISAba I* upstream of the *blaOXA*-type genes, allow *A. baumannii* strains to transmit the antibiotic resistance (AR) gene to other bacteria and the environment through horizontal gene transfer (*Bahador et al., 2015*). Carbapenem-resistant *A. baumannii* (CRAB) is a top-priority pathogen, for which new antibiotics or combination therapies are desperately needed (*Shlaes and Bradford, 2018*). Lack of an efficacious treatment for MRAB and CRAB infections has caused a reversion to initial treatments involving drugs that are considered dated now, such as colistin and polymyxin B (PMB), despite the adverse effects of these drugs (*Velkov et al., 2013*).

Polymyxins are cationic polypeptide antibiotics that specifically kill Gram-negative bacteria by binding to the bacterial outer membrane (OM). These last-resort antimicrobial agents interact with the negatively charged lipopolysaccharide (LPS) components of the OM by displacing positively charged ions, such as $Mg^{2+}$ and $Ca^{2+}$. Consequently, the integrity of the cell membrane is impaired,

**eLife digest** Wrapped in a thick, protective outer membrane, *Acinetobacter baumannii* bacteria can sometimes cause serious infections when they find their way into human lungs and urinary tracts. Antibiotics are increasingly ineffective against this threat, which forces physicians to resort to polymyxin B, an old, positively-charged drug that 'sticks' to the negatively-charged proteins and fatty components at the surface of *A. baumannii*.

Scientists have noticed that when bacteria are exposed to lethal drugs, they often react by releasing vesicles, small 'sacs' made of pieces of the outer membranes which can contain DNA or enzymes. How this strategy protects the cells against antibiotics such as polymyxin B remains poorly understood.

To investigate this question, Park et al. examined different strains of *A. baumannii*, showing that bacteria resistant to polymyxin B had lower levels of outer membrane proteins but would release more vesicles. Adding vesicles from resistant strains to non-resistant *A. baumannii* cultures helped cells to survive the drugs. In fact, this protective effect extended to other species, shielding whole communities of bacteria against polymyxin B. In vivo, the vesicles protected bacteria in moth larvae infected with *A. baumannii*, leading to a higher death rate in the animals. Experiments showed that the negatively-charged vesicles worked as decoys, trapping the positively-charged polymyxin B away from its target.

Taken together, the findings by Park et al. highlight a new strategy that allows certain strains of bacteria to protect themselves from antibiotics, while also benefitting the rest of the microbial community.

leading to cell death (*Jeannot et al., 2017*; *Moffatt et al., 2019*). Modification of lipid A in LPS by *pmrCAB* and *arnBCADTEF* gene products, which are modulators of the bacterial OM surface charge and permeability, is generally recognized as a primary mechanism of resistance to polymyxins (*Trent et al., 2001*; *Olaitan et al., 2014*; *Jeannot et al., 2017*). The *pmrCAB* and *arnBCADTEF* operons regulated by a PmrAB two-component system modify lipid A by adding 4-amino-4-deoxy-L-arabinose (LAra4N) and phosphoethanolamine (PetN), respectively (*Kline et al., 2008*; *Arroyo et al., 2011*; *Pelletier et al., 2013*). Activation of the *arnT* operon (*arnBCADTEF*, also called as the *pmrHFIJKLM* operon) adds LAra4N groups to lipid A only in several Gram-negative bacteria such as *Escherichia coli, Yersinia pestis, Pseudomonas aeruginosa*, and *Ralstonia solanacearum*, but not in *A. baumannii* or *Citrobacter rodentium* (*Trent et al., 2001*; *Sinha et al., 2019*). Because *A. baumannii* lacks the genes for the biosynthesis of LAra4N, it uses PetN addition as the main poly-myxin-resistance mechanism (*Gerson et al., 2019*). In PMB-resistant *A. baumannii*, the mutant PmrB protein is autophosphorylated, consequently phosphorylating the cytoplasmic transcriptional regula-tor PmrA. Phosphorylated PmrA then upregulates the PetN transferase PmrC and prevents PMB binding to lipid A (*Cannatelli et al., 2014*). These modifications in lipid A decrease the negative charge of the cell membrane, reducing the membrane-binding affinity of PMB. Consequently, the cell is protected from the PMB-mediated disruption of its membrane and survives (*Trimble et al., 2016*). In some *A. baumannii* clinical isolates, expression of the *eptA* gene is activated by the adja-cent *ISAba1* sequence, and the gene product PmrC homolog contributes to PMB resistance (*Trebosc et al., 2019*). Interestingly, *E. coli* producing outer membrane vesicles (OMVs) can survive in the presence of PMB (*Manning and Kuehn, 2011*; *Kulkarni et al., 2015*; *Kim et al., 2018*).

Gram-negative bacteria release OMVs with diameters of 20–400 nm to their environments. These vesicles are composed of phospholipids (PLs), outer membrane proteins (OMPs), and LPSs or lipooli-gosaccharides (LOS), but also contain periplasmic proteins and cell wall components (*Roier et al., 2016a*). Modulation of peptidoglycan (PG) cross-links is associated with membrane stability and rigidity (*Schwechheimer and Kuehn, 2015*). Approximately one-third of the alpha-helical OM lipo-protein Lpp (also known as Braun's lipoprotein) anchored to the outer cell membrane are covalently cross-linked with PGs through connection between C-terminal lysine residue of Lpp and the diamino-pimelic acid (meso-DAP) site of PG (*Dramsi et al., 2008*). Reduction in Lpp production disrupts the cross-linkage of PG on the membrane, thereby significantly contributing to the formation of OMVs (*Wessel et al., 2013*). Differential expression of OM protein A (OmpA), which is a β-barrel porin that

non-covalently interacts with the meso-DAP site of PG, can modulate OMV excretion (*Schwechheimer and Kuehn, 2015*). Many mutant studies have shown that the absence of inner membrane (IM) proteins, such as the Tol component of the Tol-Pal system and NlpA, can enhance OMV production (*Bernadac et al., 1998*; *Toyofuku et al., 2015*). Production of OMVs is also facilitated by a high periplasmic turgor pressure, which occurs when misfolded proteins, nucleotide fragments, or PG debris accumulate in the periplasm (*Brown et al., 2015*). Additionally, in *Haemophilus influenzae*, OMVs are also released by PLs that accumulate on the OM when the PL transporters of the VacJ/Yrb ABC transport system (also known as the Mla system) are not produced (*Roier et al., 2016b*). Changes in membrane fluidity and rigidity, determined by the compositions of acyl chains and head groups of PLs, affect OMV production (*Mashburn-Warren et al., 2008*). The proportion of saturated fatty acids appears to be higher in OMVs than in cellular membranes (*Baumgarten et al., 2012*).

In pathogenic bacteria, these spherical OMVs formed by various routes can have many different physiological functions, including biofilm formation, cell–cell communication, secretion of virulence factors, and survival under antibiotic stress (*Toyofuku et al., 2015*; *Bonnington and Kuehn, 2016*). OMVs have attracted interest as putative vaccine agents or vehicles for drug delivery (*Jan, 2017*). They can reduce the effectiveness of antibiotics and other bactericidal agents, thereby posing a serious threat to the treatment of bacterial infections (*Koeppen et al., 2016*; *Jan, 2017*). For example, the OMVs of β-lactam-resistant *E. coli* are enriched with proteins involved in degrading β-lactam antibiotics in comparison with those of susceptible bacteria (*Kim et al., 2018*). Accordingly, OMV supplementation of microbial cultures increases the microbial resistance to antimicrobial peptides. Similarly, hypervesiculating mutants show increased resistance (*Manning and Kuehn, 2011*). OMVs from *A. baumannii* are known to possess various virulence factors such as OmpA, proteases, phospholipases, superoxide dismutase, and catalase (*Kwon et al., 2009*). However, it is unclear whether *A. baumannii* can gain polymyxin resistance or confer such resistance onto its bacterial community through hypervesiculation. Here, the genomes and OMV-producing phenotypes of experimental evolution PMB-resistant *A. baumannii* cells were analyzed alongside other clinical isolates as well as the *A. baumannii* ATCC 17978 strain. Our multi-omics data and mutational studies suggested that OMVs functioned as decoys for PMB in *A. baumannii.* Accordingly, these vesicles can protect not only the OMV producer, but also the entire bacterial community from the bactericidal effects of PMB.

## Results

### OMV secretion and enhanced biofilm formation

PMB-resistant strains were generated by exposing wild-type cells (Lab-WT; minimum inhibitory concentration [MIC], 2 μg/mL) to increasing concentrations of PMB. Consequently, strains that had 2-fold (MIC, 4 μg/mL) and 64-fold (MIC, 128 μg/mL) higher PMB resistance than Lab-WT were generated (low and high PMB resistance, PMR$^{Low}$ and PMR$^{High}$, respectively) (*Figure 1A*, *Figure 1—source data 1*). The PMR$^{High}$ strain had a lower growth rate ($1.386 \pm 0.17$ h$^{-1}$) than both Lab-WT ($2.081 \pm 0.92$ h$^{-1}$) and PMR$^{Low}$ ($1.985 \pm 0.86$ h$^{-1}$) strains (*Figure 1—figure supplement 1A*). Expression analyses of the *pmrC* gene, which encodes a PetN transferase in the absence of PMB, showed 1.2- and 4-fold higher expression in PMR$^{Low}$ and PMR$^{High}$ strains, respectively (*Figure 1—figure supplement 1B*). The zeta-potential analyses suggested that the average negative surface charge of PMR$^{Low}$ and PMR$^{High}$ decreased from −27 mV (Lab-WT) to −20 mV and −10 mV, respectively (*Figure 1B*, *Figure 1—source data 1*). The reduced surface charge, which is partly due to PmrC-driven incorporation of PetN into lipid A, presumably decreases the initial binding of PMB, a cationic amphiphilic cyclic decapeptide, to the cell surface. Consequently, cells with more PetNs in lipid A acquire resistance to PMB. PmrC-mediated reduction of surface charges was not feasible in PMR$^{Low}$ because the expression level of the *pmrC* gene was not significant. These observations could not account for all the reasons underlying the high PMB resistance of PMR$^{High}$ strain because short-term (10 min) PMB exposure (4 μg/mL, MIC of PMR$^{Low}$) decreased the surface charge in all tested strains, albeit to different extents (*Figure 1—figure supplement 1C*, n = 10 per strain). OMV-associated biofilm was measured in the Lab-WT and PMR$^{High}$ strains (*Figure 1—figure supplement 1D, E*). Biofilm formation observed using confocal microscopy with FilmTracer indicated that the PMR$^{High}$ strain

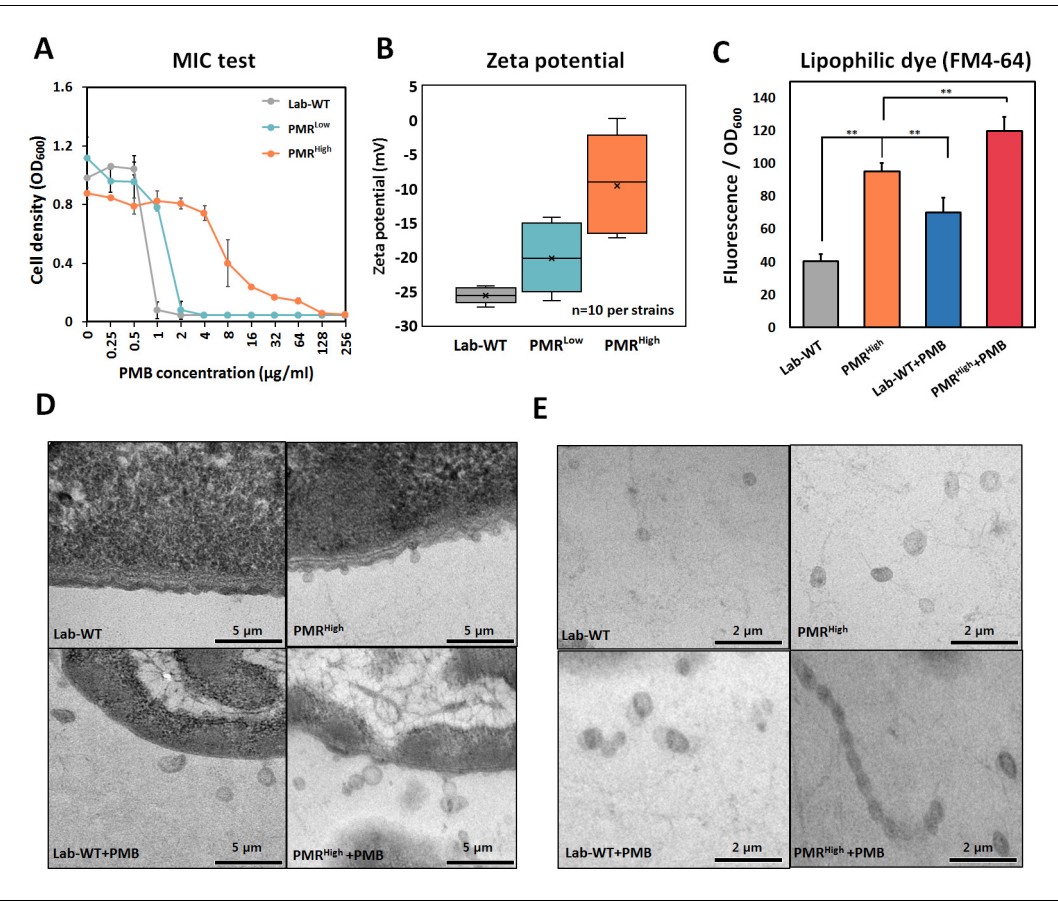

**Figure 1.** Generation of experimentally evolved polymyxin B (PMB)-resistant strains and observation of outer membrane vesicle (OMV) production. (**A**) PMB-resistant strains generated by exposing wild-type cells to increasing concentrations of PMB (N = 3 biological replicates). (**B**) Zeta-potential of the Lab-WT, PMR$^{Low}$, and PMR$^{High}$ strains (n = 10 each) under the non-treated condition. (**C**) OMV production was quantified by using a lipophilic dye (FM4-64) under the non-treated condition or 1/2 minimum inhibitory concentration of PMB treatment condition (N = 3 biological replicates). Asterisks indicate statistical significance at p<0.01. (**D**) Transmission electron micrographs (TEMs) of the experimentally evolved strains. Cells were grown to the exponential phase in LB liquid medium. (**E**) Detached OMVs from each strain were observed using TEM. OMVs were connected in a row in the PMR$^{High}$ strain. The online version of this article includes the following source data and figure supplement(s) for figure 1:

**Source data 1.** The minimum inhibitory concentration of experimentally evolved polymyxin B-resistant strains and outer membrane vesicle production.
**Figure supplement 1.** Growth, membrane charges, and biofilm formation of the experimentally evolved strains.
**Figure supplement 2.** Measurement of minimal inhibitory concentrations (MICs) in Lab-WT and polymyxin B-resistant cells (PMR$^{Low}$ and PMR$^{High}$).

produced thick and condensed biofilm within 24 hr, and the addition of PMB induced biofilm formation in both strains (*Figure 1—figure supplement 1D*). Quantification of biofilm formation using the crystal violet assay also indicated that the PMR$^{High}$ strain produced more biofilm (twofold higher) than the Lab-WT strain (*Figure 1—figure supplement 1E*). Interestingly, addition of OMVs derived from PMR$^{High}$ increased the initial stage of Lab-WT biofilm formation (*Figure 1—figure supplement 1F, G*). The PMR$^{Low}$ and PMR$^{High}$ strains were resistant only to polymyxins, but not to the other nine tested antibiotics of different classes, including meropenem, rifampicin, chloramphenicol, and ampicillin (*Figure 1—figure supplement 2*). This PMB-specific phenotype prompted us to confirm hypervesiculation of the PMR-resistant strains through transmission electron microscopy (TEM). OMV production appeared to be twofold higher in PMR$^{High}$ than in Lab-WT, and the treatment with 1/2 MIC PMB (1 or 128 μg/mL, respectively) caused increased OMV production as quantified using the lipophilic dye FM4-64 (*Figure 1C*). We observed markedly larger and greater production of OMVs in

the PMR$^{High}$ strain than in our Lab-WT strain using TEM (*Figure 1D*, *Figure 1—figure supplement 1H*). Dissociated OMVs from PMR$^{High}$ were larger and became entangled with other OMVs through hairy structured materials (*Figure 1E*).

## Analyses of modified lipid A and Raman spectroscopy

To determine whether modified lipid A plays an important role in PMB resistance, lipid A was subjected to matrix-assisted laser desorption/ionization time-of-flight (MALDI-TOF) analysis in the positive-ion mode. The lipid A extraction method is summarized in Materials and methods. The main peak of Lab-WT was measured at 1490 m/z, corresponding to bis-phosphorylated penta-acylated lipid A (*Figure 2A, B*). In the PMR$^{High}$ strain, the predominant spectrum was also measured at 1490 m/z, followed by 1630 m/z, which is predicted to be the peak when a PetN and a hydroxyl group were attached to the dominant lipid A (*Figure 2A, B*). The PMR$^{High}$ strain overexpressed *pmrC* (encoding a PetN transferase), which added a PetN to some lipid A molecules, but not all, leading to interference PMB binding to the OM (see *Figure 1—figure supplement 1B*). Raman spectroscopy is used for microbial analysis at a single-cell level since it can rapidly measure the cell components, including lipids, nucleic acids, and proteins, by detecting the inelastic scattering of a molecule irradiated by monochromatic lights. The phenylalanine peak (1001–1004 cm$^{-1}$) appears to be a signature signal for the presence of peptides and proteins in a sample. This phenylalanine peak at 1002 cm$^{-1}$ appeared to be similar in both strains (*Figure 2C*). However, the Raman peak corresponding to PL (at 1334, 1445, and 1666 cm$^{-1}$) increased notably in PMR$^{High}$ (*Figure 2C*). The intensity of the other prominent peak at 1243 cm$^{-1}$ assigned to an amide III β-sheet was higher in resistant strains (*Figure 2C*). Increased magnitude of Raman intensity at amide III, along with hydroxyl, carboxyl, and phosphoryl groups, was reported in bacterial biofilm, indicating profound changes in the bacterial cell membrane (*Jung et al., 2014*; *Kim et al., 2019*). The Raman peaks of amide III were increased in clinical quinolone-resistant *E. coli* (*Kim et al., 2019*). Principal coordinate analysis (PCoA) using the collected Raman spectra of bacterial cells revealed that dispersion of individual dots was clustered into two groups, with 38.2% explanatory power, although some dots were in shared areas between Lab-WT and PMR$^{High}$ (*Figure 2D*). Individual cell heterogeneity analysis showed that Lab-WT strains were more heterogeneous than resistant strains, suggesting that PMR$^{High}$ strains were specifically adapted and unified by PMB stress (*Figure 2E*).

## Integration of genomic, transcriptomic, and proteomic analyses

We hypothesized that the higher production of OMVs secreted by the PMR$^{High}$ strain functioned as a decoy target for PMB. In this scenario, OMVs are expected to protect their mother cells as well as the entire bacterial community from the effect of PMB. To gain a better insight into the mechanism underlying PMB resistance and OMV biogenesis in the PMR$^{High}$ strain, whole-genome sequencing (WGS) analyses of PMR$^{Low}$ and PMR$^{High}$ were conducted (*Supplementary file 1b*). It has been known that *A. baumannii* ATCC 17978 strain possesses a single chromosome and two plasmids, namely pAB1 (13,408 bp) and pAB2 (11,302 bp). The same small plasmid, pAB2 (11,299 bp, three bases were deleted), was detected only in the PMR$^{High}$. The large plasmid was missing in both evolved strains (*Figure 3—figure supplement 1A*). The chromosome sizes of ATCC 17978, PMR$^{Low}$, and PMR$^{High}$ were 3,976,747 bp, 3,971,618 bp, and 3,955,017 bp, respectively (*Figure 3—figure supplement 1A*). Gene alignments and comparative genomic analyses were performed on the reference ATCC 17978 strain (Lab-WT). The WGS data were assembled to construct draft genomes with median N50 values of 3971 kb for PMR$^{Low}$ and 3955 kb for PMR$^{High}$ (*Supplementary file 1b*). The numbers of coding DNA sequences (CDSs) in the PMR$^{Low}$ and PMR$^{High}$ genomes were 3678 and 3762, with mean CDS lengths of 947.5 and 930.2, respectively (*Supplementary file 1b*). Both MUMmer program-based genomic and additional PCR analyses were performed to confirm mutational positions (*Supplementary file 1a, b*). No mutation was detected in the promoter (all intergenic) regions of either mutant strain genome. Two chromosomal regions mostly annotated as hypothetical proteins were lost during adaptive evolution of PMR$^{Low}$ and PMR$^{High}$ strains (*Supplementary file 1c*). All 20 antibiotic-resistant genes and 78 IS elements remain unchanged in both evolved strains (*Supplementary file 1d, e*). We focused on PMR$^{High}$ rather than PMR$^{Low}$ because the first strain produced more OMVs than the latter. In the PMR$^{High}$ strain, mutations were found in the protein damage repair gene *surE*, fimbrial gene *fimT*, cell division-related gene *ftsL* (nonsense mutation), DNA

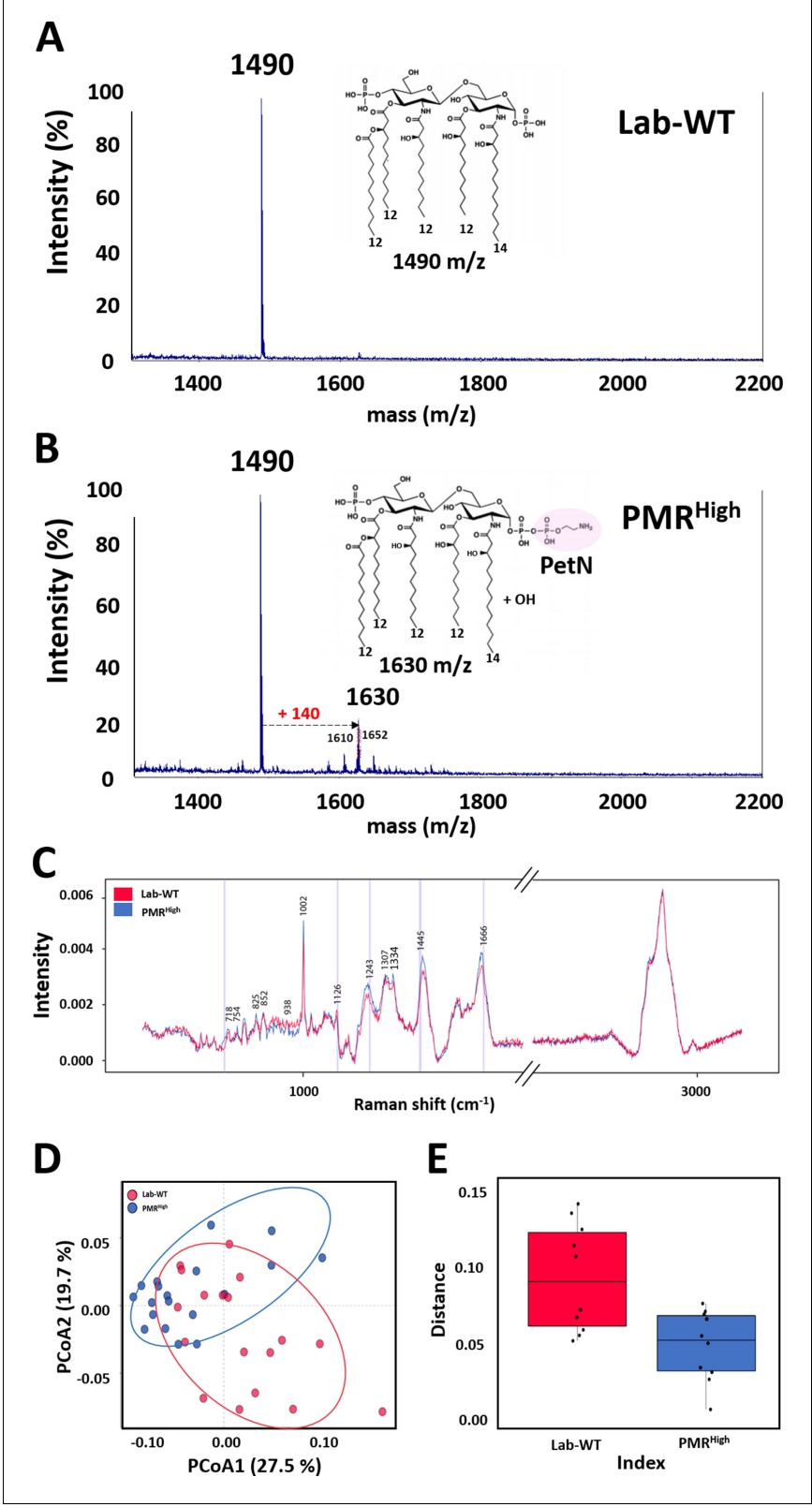

**Figure 2.** Mass spectrometry and proposed structures of lipid A isolated from Lab-WT and PMR[High]. (**A**) Lab-WT produced peaks at an m/z value of 1490, corresponding to bis-phosphorylated penta-acylated lipid A. (**B**) PMR[High] produced peaks at m/z values of 1490 and 1630, corresponding to bis-phosphorylated penta-acylated lipid A without/with one phosphoethanolamine group and -OH, respectively. (**C**) Raman spectra of single *A. baumannii*

*Figure 2 continued on next page*

*Figure 2 continued*

cells grown to the exponential phase in LB media. 15–20 single spectra were used to produce mean spectra. (D) Principal coordinate analysis (PCoA) demonstrated separation between Raman spectra of Lab-WT and PMR$^{High}$. (E) Heterogeneity of the PCoA dots was analyzed.

repair gene *udg,* DNA integration gene *xerD*, and fatty acid synthesis-related gene *fabH* (nonsense mutation) (*Supplementary file 1a*, *Figure 3—figure supplement 1A*). Additionally, comparative genomic analysis between Lab-WT and PMR$^{High}$ strains revealed point mutations in the *pmrB* gene of the PMR$^{High}$ strain. PmrB was the protein that showed different mutations in the PMR$^{Low}$ and PMR$^{High}$ strains, with amino acid substitutions at N353I and T2351, respectively (*Supplementary file 1a*, *Figure 3—figure supplement 1A*). The mutations in N353I and T2351 in PmrB are known to increase the expression of the *pmrC*, resulting in PetN modification of lipid A (*Adams et al., 2009*).

Due to high MIC under PMB and hypervesiculation of the PMR$^{High}$ strain, we focused on comparative transcriptome between lab WT and PMR$^{High}$ strain (*Supplementary file 1f*). Our RNA-seq analysis showed that 250 and 166 genes were up- and downregulated, respectively, in PMR$^{High}$ in comparison with the Lab-WT strain without PMB exposure (*Figure 3—figure supplement 1B*). In addition, 89.16% of all genes showed no change at the transcriptional level. The highly upregulated gene, encoding an aldehyde dehydrogenase (*aldB*), showed a 3.43-fold increase in the PMR$^{High}$ in comparison with that in the control (*Figure 3—figure supplement 1B*). It is worth noting that dynamic membrane lipid biosynthesis in the PMR$^{High}$ might generate long-chain fatty aldehydes that may be recycled through AldB (*Asperger and Kleber, 1991*; *Ishige et al., 2000*). 8 of the 16 most upregulated genes were hypothetical proteins, and the remaining genes were annotated as a glutamin-(asparagin)-ase (*ansB*), poly-beta-1,6-N-acetyl-D-glucosamine N-deacetylase (*pgaB*), ribosomal protein (*rpmI*), and an ABC transporter glutamine-binding protein (*glnH*) (*Figure 3—figure supplement 1B*). In particular, the PgaB belonging to the *pgaABCD* operon functions for exopolysaccharide export across the OM. The overexpressed *pga* operon appeared to facilitate the production of PNAG-attached OMVs in *E. coli* through deacetylation of PNAG by the PgaB activity, which resulted in enhanced biofilm formation (*Stevenson et al., 2018*). The PMR$^{High}$ strain showed upregulation of both *pmrA* and *pmrC* genes encoding a transcriptional regulator and a PetN transferase, respectively, which evidently affects the membrane charge leading to PMB resistance (*Figure 3—figure supplement 1B*). Notably, the most downregulated genes except the hypothetical gene were related to two genes encoding the OMPs *ompA* and *bamE* (*Figure 3—figure supplement 1B*). OmpA, a PG-linked non-specific OM porin, harbors a flexible periplasmic domain that is strongly associated with PG (*Iyer et al., 2018*). Lack of OmpA in the OM is known to promote bacterial OMV production (*Schwechheimer et al., 2014*). The *bamE*, encoding a β-barrel assembly machine E, was shown to bind specifically to phosphatidylglycerol (PG) in the OM (*Knowles et al., 2011*). Both OmpA and BamE are determinants for cell shape and OM integrity (*Ryan et al., 2010*; *Choi and Lee, 2019*). Downregulation of both genes may lead to loosening of OM-PG linkages and is associated with hypervesiculation in the PMR$^{High}$. An additional qRT-PCR assay confirmed that the *pmr* and *pgaB* genes showed more than fourfold higher expression, but all membrane-linkage-related genes (*ompA*, *bamE*, *lpp*, and *mlaC*) were downregulated in PMR$^{High}$ (*Figure 3A*, *Figure 3—source data 1*). Biogenesis of OMV in the PMR$^{High}$ was twice as much as that in Lab-WT and further increased under 1/2 MIC PMB (*Figure 1C*). To identify the commonly expressed genes in the high OMV production condition, we compared three conditions (Lab-WT + PMB vs. Lab-WT), (PMR$^{High}$ + PMB vs. PMR$^{High}$), and (PMR$^{High}$ vs. Lab-WT) (*Figure 3—figure supplement 2*). Our RNA-seq analyses confirmed that 29 and 12 genes were up- and downregulated, respectively, in the hypervesiculation conditions (*Figure 3—figure supplement 2A*). High levels of gene expression involved in stress response and survival were also predominant in the transcriptomic analysis, in which ribosomal proteins (*rplW* and *rplO*) and chaperons (*groL*) were upregulated in the hypervesiculation conditions (*Supplementary file 1g*). The *pgpA* gene encoding a lipid phosphatase was also upregulated, which can dephosphorylate phosphatidylglycerophosphate to form PG, a major constituent in OMVs (*Supplementary file 1g*, *Lu et al., 2011*). Increased levels of PG in the OM induced changes in membrane integrity, causing higher OMV production (*Sohlenkamp and Geiger, 2016*). Notably, several genes belonging to the *pmr* and *pga* operons are included among the commonly

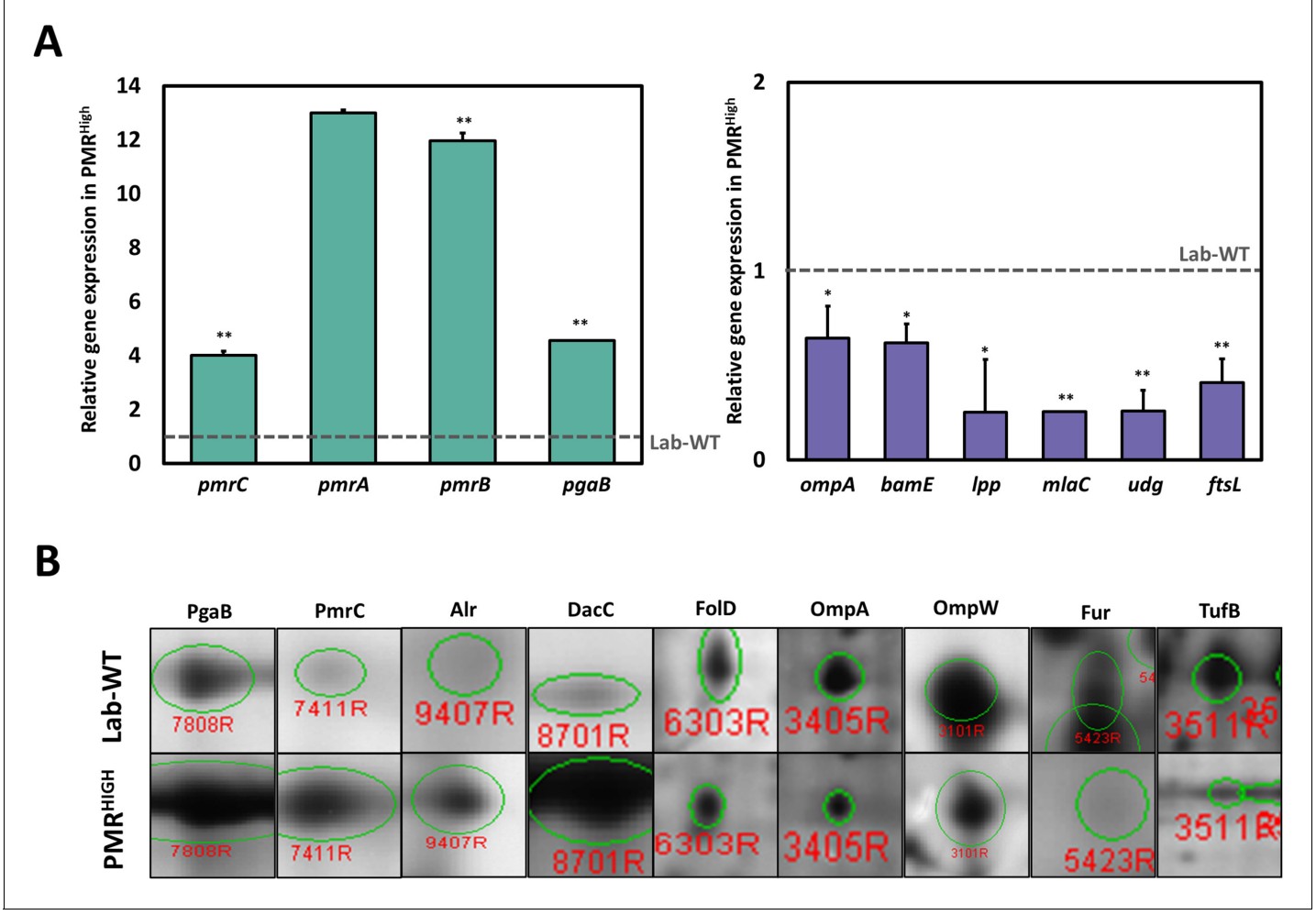

**Figure 3.** Integration of omics analysis in experimentally evolved strains and outer membrane vesicles (OMVs). (**A**) The qRT-PCR assay was conducted in PMR<sup>High</sup> strain compared with Lab-WT. The expression of the strains was normalized to their respective 16S rRNA expression. All qRT-PCR procedures were conducted in triplicate from at least three independent cultures. (**B**) Two-dimensional gel electrophoresis image analysis of proteins expressing isolated OMVs from Lab-WT and PMR<sup>High</sup> strains (*Figure 3—source data 2*). The proteins were stained by colloidal Coomassie Brilliant Blue and analyzed by Image Master Platinum 5.0 image analysis program. The selected proteins indicated up- or down-expressed proteins in Lab-WT.

The online version of this article includes the following source data and figure supplement(s) for figure 3:

**Source data 1.** The qRT-PCR assay in PMR<sup>High</sup> strain compared with Lab-WT.

**Source data 2.** Images of selected spots in proteomics analysis (Lab-WT, Lab-WT + 1/2 minimum inhibitory concentration [MIC], PMR<sup>High</sup> and PMR<sup>High</sup> + 1/2 MIC).

**Source data 3.** The data of two-dimensional gel electrophoresis analysis for identifying differentially expressed proteins in our isolated outer membrane vesicles collected from experimentally evolved strains.

**Figure supplement 1.** Genomics and transcriptomic analyses of experimentally evolved strains.

**Figure supplement 2.** Transcriptome and proteome analyses of differential conditions.

upregulated gene categories, and the commonly downregulated genes included important membrane-associated genes (*mlaC*, *ompA*, *lpp*, and *bamE*) (*Supplementary file 1g*).

Purified OMVs were separated by two-dimensional gel electrophoresis (2-DE) followed by Coomassie blue staining. Gel-based proteomics with MALDI-TOF analysis were performed under the same four conditions (*Figure 3B*, *Supplementary file 1h*, *Figure 3—source data 2* and *3*). 43 out of the 61 up- or downregulated proteins were designated with the MS-Fit program by using the NCBInr and UniPortKB databases, and 18 spots were not identified (*Supplementary file 1h*). The commonly identified OMV components harbored several proteins that were noted in our RNA-seq analyses for OMV biogenesis, showing increased production (PmrC, alanine racemase [Alr], and

PgaB) and decreased production (OmpA and OmpW, elongation factor Tu [TufB]). Both our RNA-seq and proteomics analyses strongly indicated that simultaneously identified genes and proteins are critical components of OMV biogenesis in both wild-type and PMR$^{High}$ strains. Interestingly, *A. baumannii* OMVs also contained many cytoplasmic cargo proteins such as ribosomal proteins and Ef-Tu, similar to the OMVs from other bacteria (*Dallo et al., 2012*; *Deo et al., 2018*). Elongation factor (Ef-Tu), observed in our proteomics analyses, is one of the most abundant proteins (>20%) in bacteria and has a canonical role in translation. However, it is also found on the surface of bacterial OMs, which suggests a possible noncanonical function on the membrane (*Widjaja et al., 2017*; *Harvey et al., 2019*). Accumulation of Ef-Tu in the periplasmic space might induce pressure, which promotes blebbing of the OM resulting in OMV secretion although lower level of TufB was detected in PMR$^{High}$ strain (*Figure 3B*, *McBroom and Kuehn, 2007*). Reduced expression of OmpA and OmpW leads to diminished linkage between the OM and PG and subsequently causes more OMV generation in the PMR$^{High}$. Our RNA-seq and proteomics data implied that PMR$^{High}$ changes the OM surface charge and membrane integrity by altering the expression of many OMV-related genes, which leads to PMB resistance and higher OMV production. Decreased membrane-linkage proteins induce overproduction of OMVs, which is also linked to enhanced biofilm formation.

## Mutational analyses for OMV production

OMV production and MIC were measured in all strains, including several Campbell-type and CRISPR-Cas9-based knockout strains (*Figure 4—figure supplement 1A*). We used several approaches for quantifying OMV production (lipid staining [FM4-64 dye], quantification of total proteins [Bradford assay], total DNA measurement, and LPS quantitation using the purpald assay) (*Figure 4*, *Figure 4—source data 1*, *Figure 4—figure supplement 1B*). OMV production appeared to be 1.5- to 2-fold higher in PMR$^{High}$ than PMR$^{Low}$ or Lab-WT, as quantified by using the lipophilic dye FM4-64 (*Figure 4A*, *Figure 4—source data 1*). Total protein, DNA, and LPS amounts in OMVs collected from the same amounts of cell cultures were higher in PMR$^{High}$ than Lab-WT (*Figure 4B–D*). Furthermore, when the *pmrB* gene was deleted in Lab-WT, OMV biogenesis was reduced (0.3-fold) (*Figure 4A–D*), indicating that OMV production is impaired in the absence of PmrB. This finding was supported by our observation in which an extra copy of *pmrB$^{H}$* gene increased OMV production in Lab-WT (*Figure 4A–D*). However, the Δ*ftsL*, Δ*udg*, Δ*lpp*, and Δ*mlaC* mutants exhibited higher OMV production (*Figure 4A–D*). As expected from our omics data, OMV production was reduced when the PNAG transporter-related gene, *pgaB*, was deleted in Lab-WT (*Figure 4A–D*). The relative OMV production measured by FM4-64, protein, DNA, and LPS quantification in the Δ*fabH* and Δ*surE* mutants could be ignored because no statistically significant difference was found (*Figure 4—figure supplement 1C–F*). Our mutational study clearly demonstrated that mutant strains lacking genes involved in membrane cross-linkages and PL management (*lpp*, *ftsL*, and *mlaC*) showed increased OMV secretion. The MICs of the *pmrB*-deleted strain (Δ*pmrB*) appeared to be reduced (*Figure 4E*). Introduction of the complementary *pmrB* gene to Lab-WT (Lab-WT + pRK-*pmrB*) increased the MIC, whereas the empty vector (pRK) did not affect the MIC (*Figure 4—figure supplement 1B*). The MICs of Δ*ftsL*, Δ*udg*, Δ*lpp*, and Δ*pgaB* remained unchanged in comparison to that of Lab-WT (*Figure 4E*). The MICs of all constructed mutants were not affected, although they showed higher OMV production due to exposure to PMB at the beginning stage of culture conditions, in which no OMV production was expected. However, tendency for protecting cells grown in stationary phase from high PMB concentration (1/2 MIC of each clinical strain) was observed in our tested higher OMV producers (*Figure 4F*).

## Protective effect of OMVs against PMB

We hypothesized that the OMVs secreted by PMR$^{High}$ and mutant strains act as decoys to mask the effect of PMB on cells. To assess this hypothesis, we extracted OMVs from PMR$^{High}$ culture (*Figure 5—figure supplement 1A*) and confirmed their presence using flow cytometry (*Figure 5—figure supplement 1B*). Concentrated OMVs were negatively stained with uranyl acetate and visualized using TEM (*Figure 5—figure supplement 1C*). The protective effect of these PMR$^{High}$-derived OMVs was assessed using the Luria–Bertani (LB) broth (*Figure 5A*, *Figure 5—source data 1*) and LB agar plate tests (*Figure 5B*), after confirming that additional purified OMVs had no effect on the bacterial growth. Interestingly, supplementation of PMB-containing liquid cultures (*Figure 5A*) or

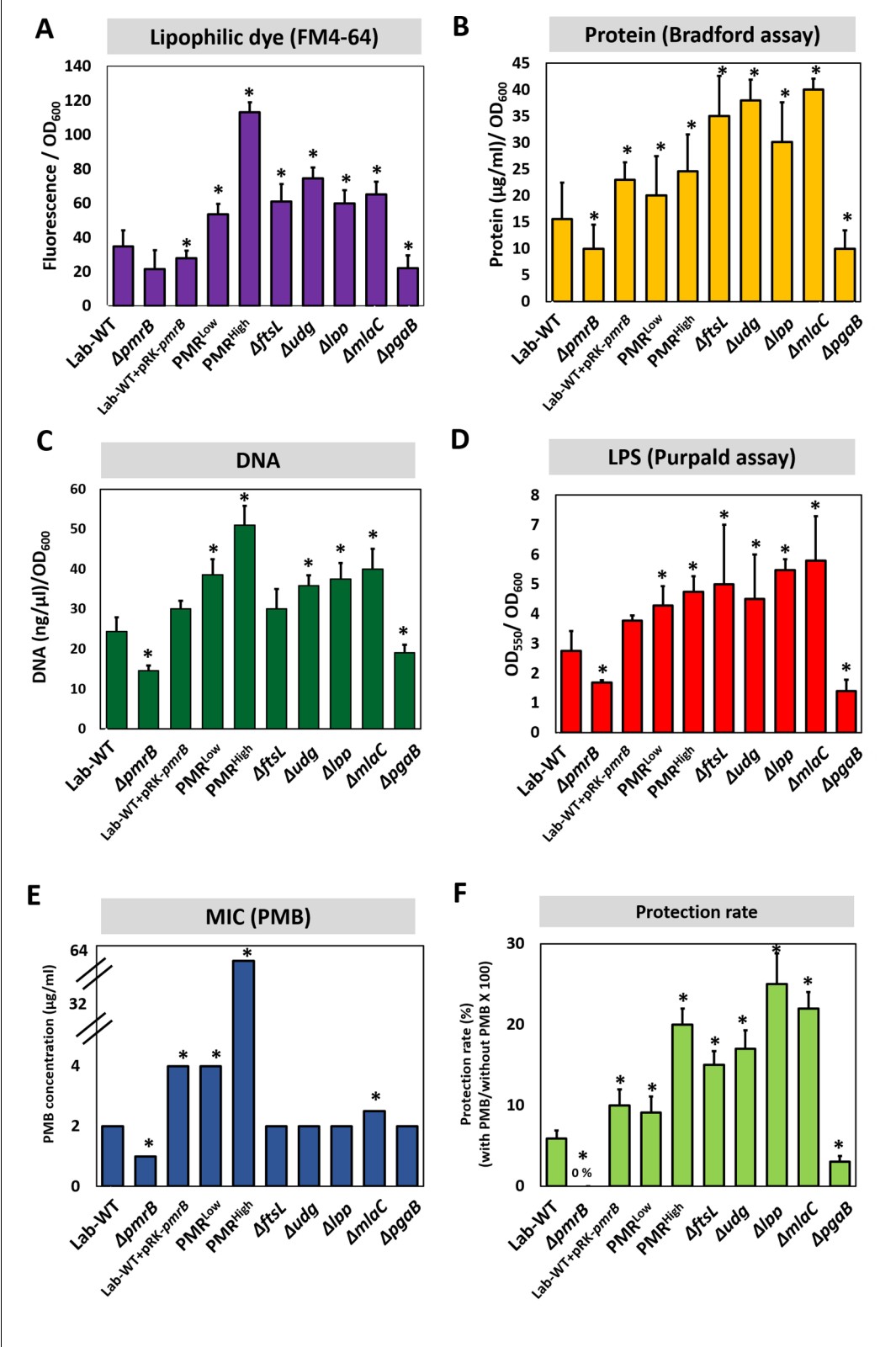

**Figure 4.** Polymyxin B (PMB) susceptibility test and outer membrane vesicle (OMV) production in experimentally evolved stains and mutants. OMV production was analyzed for (A) lipids (lipophilic dye), (B) total protein (Bradford assay), (C) total DNA amount, and (D) lipopolysaccharide amount (purpald assay) in the exponential phase. Each measurement was normalized by the equivalent $OD_{600}$. (E) The minimum inhibitory concentration (MIC)

*Figure 4 continued on next page*

Figure 4 continued

of PMB in each strain was tested. (F) Tendency for protecting cells grown in stationary phase from high PMB concentration (1/2 MIC of each clinical strain). No protection was observed in ΔpmrB. In all panels, an asterisk indicates statistical significance at p<0.05 (N = 3 biological replicates).

The online version of this article includes the following source data and figure supplement(s) for figure 4:

**Source data 1.** Mutational analysis and quantification of outer membrane vesicle in mutants.
**Figure supplement 1.** Generation of knockout mutants (ΔfabH and ΔsurE) and their outer membrane vesicle (OMV) production.

agar (**Figure 5B**) with these OMVs could increase the survival of all tested strains except for PMR[High] on agar plates. To visualize direct binding PMB to OMVs, we developed a dansyl fluorophore-PMB, which was synthesized by forming a chemical bond between the primary γ-amines on the diaminobu-tyric-acid residues of PMB and dansyl-chloride (**Soon et al., 2011**). Confocal laser scanning micros-copy (CLSM) images revealed that dansyl-PMB was less bound to PMR[High] cells than Lab-WT cells at their respective PMB MICs (2 or 4 µg/mL) due to the fact that the PmrC-driven lipid A modification of PMR[High] cells decreased PMB binding to OMs (**Figure 5C**). The lipid-selective dye (FM4-64) and dansyl-PMB were employed to visualize binding of PMB to purified OMVs of the PMR[High] stain (**Figure 5D**). Our data showed that the two dyes exhibited overlapping images, which clearly dem-onstrated the direct binding of PMB to OMVs. To test whether *A. baumannii* clinical isolates could be protected from PMB through OMV secretion, the linkages between PMB protection rates and

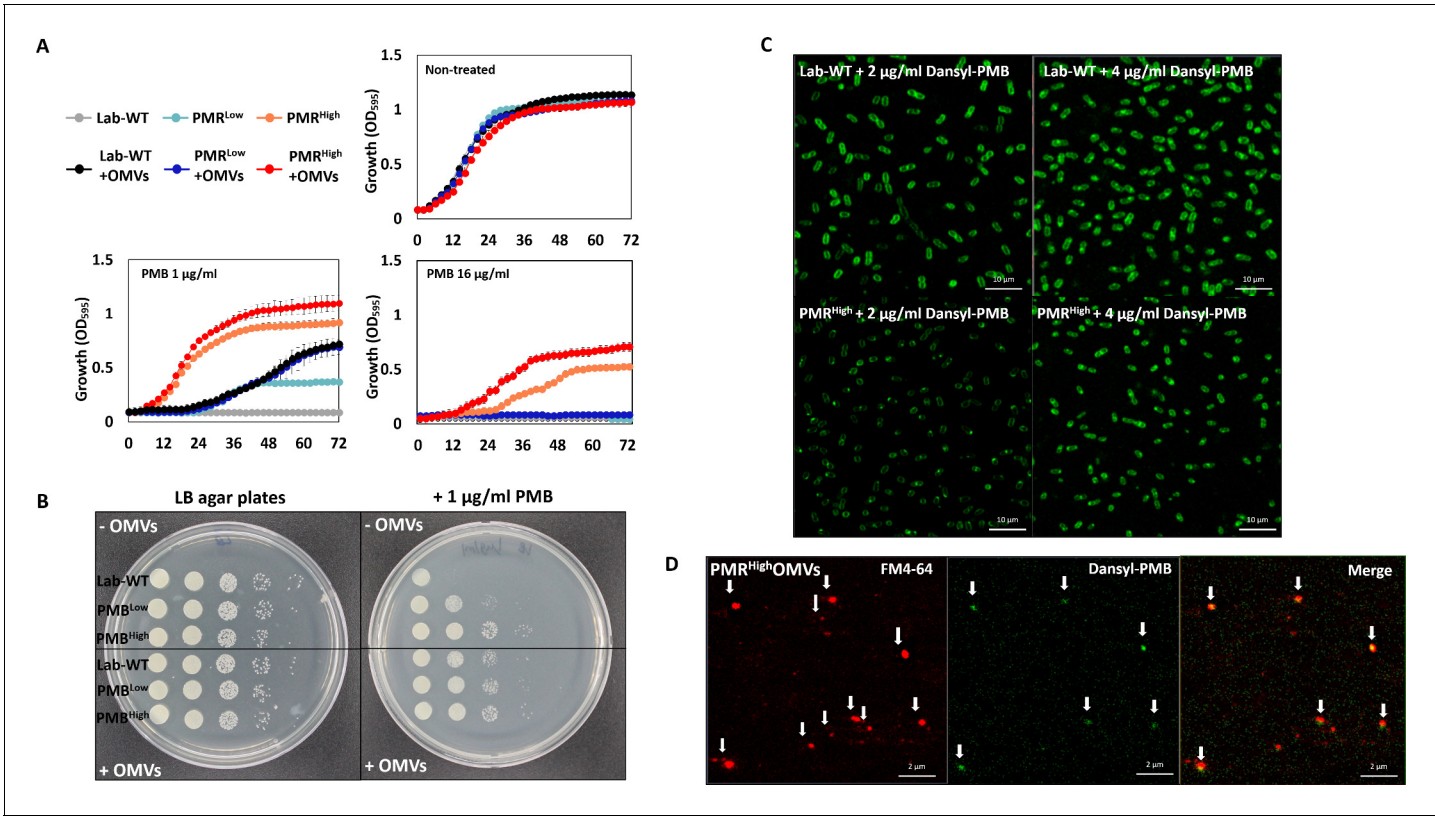

**Figure 5.** Protective effects of outer membrane vesicles (OMVs) in *A. baumannii* measured in vitro and visualization of the decoy role of OMVs. (A) The results are shown after 72 hr. The Lab-WT, PMR[Low], and PMR[High] strains were used to measure the protective effects of OMVs (N = 3 biological replicates). (B) Protective effects were measured after the addition of cell suspensions in LB medium with/without OMVs to polymyxin B (PMB)-containing agar. Images were obtained after 14 hr of incubation at 37°C. (C) Confocal laser scanning microscopy images of Lab-WT and PMR[High] treated with dansyl-PMB for 30 min at 37°C. (D) Isolated OMVs from PMR[High] were stained with FM4-64 and dansyl-PMB.

The online version of this article includes the following source data and figure supplement(s) for figure 5:

**Source data 1.** The Lab-WT, PMR[Low], and PMR[High] strains were used to measure the protective effects of outer membrane vesicles after 72 hr.
**Figure supplement 1.** Procedures for collecting outer membrane vesicles (OMVs) and verifying extracted OMVs using fluorescence-activated cell sorting (FACS) and transmission electron microscopy (TEM).

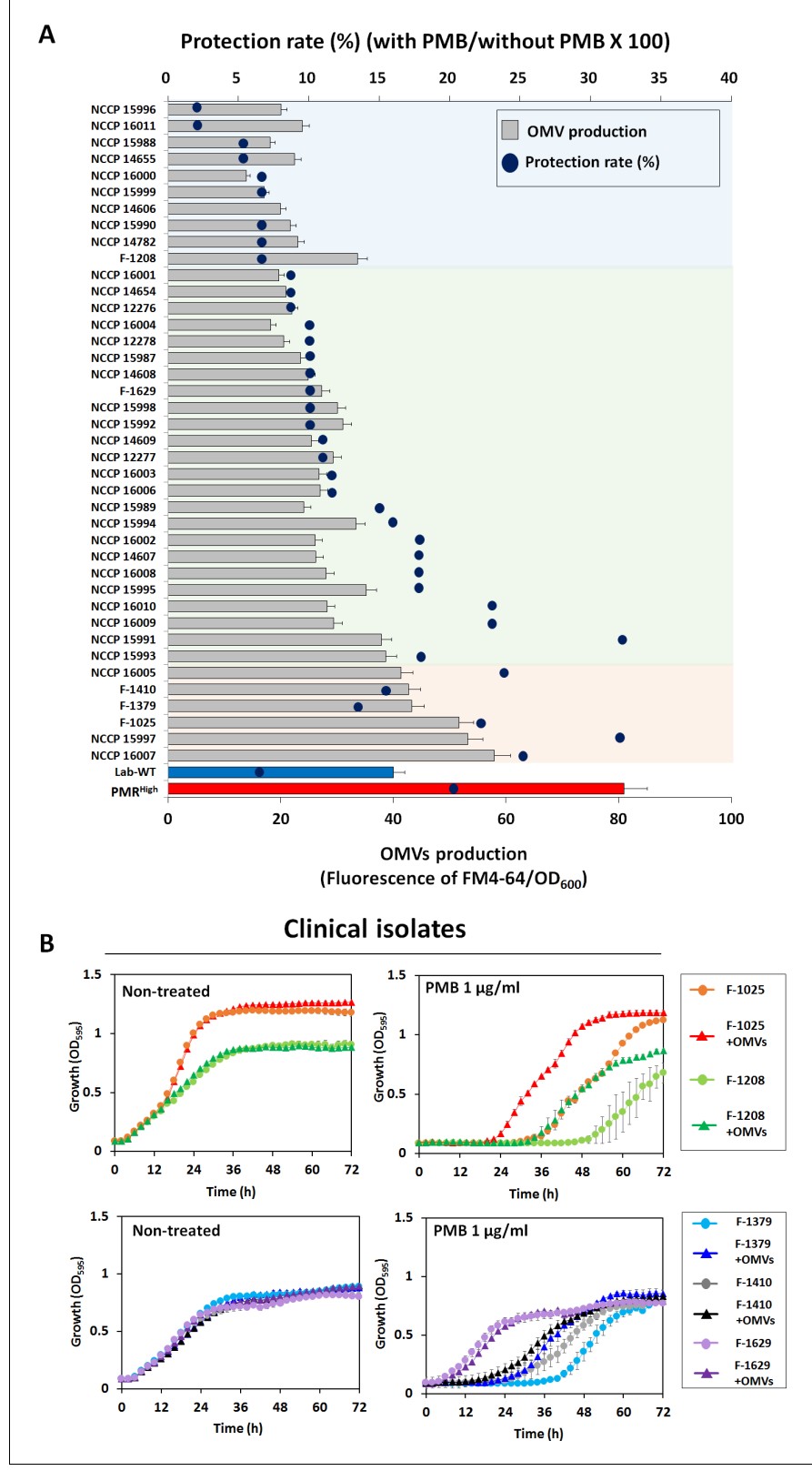

**Figure 6.** Outer membrane vesicle (OMV) production and protective effects in clinical isolates. (**A**) Minimum inhibitory concentration value and OMV production were examined in 40 MDR clinical *A. baumannii* strains. Blue dots show the protection rate for treatment of 1/2 MIC polymyxin B (PMB) in each clinical isolate, and the bar graph shows the amount of OMVs in the early stationary phase ($OD_{600} = 0.4$). The colored area indicates degree

*Figure 6 continued on next page*

*Figure 6 continued*

of correlation between OMV production and protection rate in stationary phase-grown cells. Red area: strong correlation; green area: moderate correlation (N = 3 biological replicates). (B) Protective effects of OMVs in five clinical *A. baumannii* strains (F-1025, F-1208, F-1379, F-1410, and F-1629) measured using LB liquid medium without and with 1 μg/mL PMB after treatment with OMVs. The $OD_{600}$ of the strains was measured in the wells of 96-well plates until 72 hr of incubation at 30°C (N = 3 biological replicates).

The online version of this article includes the following source data and figure supplement(s) for figure 6:

**Source data 1.** Outer membrane vesicle production and protective effects in clinical isolates.

**Figure supplement 1.** In vitro protective roles of outer membrane vesicles (OMVs) in other Gram-negative bacteria.

OMV production in 40 MDR clinical isolates were assessed (*Figure 6A*, *Figure 6—source data 1*). The degree of OMV production in each clinical isolate varied. Although direct correlations between the MIC and OMV production could not be identified under our tested conditions due to the same reasons we provided earlier, higher OMV producers could show higher protection rates against PMB in the early stationary phase (*Figure 6A*). Particularly, tendency was greater in the six clinical isolates with red-colored area (*Figure 6A*). Interestingly, addition of PMR$^{High}$-derived OMVs to the growth cultures of five randomly chosen MDR strains (*Supplementary file 1i*; F-1025, F-1208, F-1379, F-1410, and F-1629) enhanced the microbial survival in the presence of PMB (*Figure 6B*). PMB MIC tests were performed using four additional Gram-negative bacteria (*P. aeruginosa* PAO1, *Pseudomonas putida* KT2440, *E. coli* K12, and *Acinetobacter oleivorans* DR1) and the protective roles of their OMVs were tested (*Figure 6—figure supplement 1*). Our data suggested that OMV-driven protection under PMB conditions could be applicable to other Gram-negative bacteria.

## Protective effects of OMVs in in vitro human gut microbiota

We next tested whether OMVs could mask the effect of PMB on the entire bacterial microbiota. To this end, we used human fecal microbiota and employed the survival of *Galleria mellonella* as an infection model (*Figure 7*). The fecal microbiota was incubated for 5 days under anaerobic conditions to verify the extended protective effect of OMVs. Cultivation was continuously performed after dividing the sample into three groups (a control group without PMB, a second bottle with PMB, and a third bottle with PMB after addition of OMVs 3 hr earlier). PMB and OMVs were added daily for 5 days. The environmental MIC of PMB was determined and 1/2 environmental MIC was used in the following experiments (~50% reduction of aerobic bacterial microbiota in 24 hr) (*Figure 7—figure supplement 1A*). After 5 days, the community cells were spotted on an LB plate and cultured under aerobic conditions to check PMB toxicity and the protective effect of OMVs (*Figure 7—figure supplement 1B*). The PMB-treated sample showed a $10^4$-fold reduction in aerobic bacterial cells than the control, and the OMV plus PMB sample showed similar levels of cell counts as the control (*Figure 7—figure supplement 1B*).

Bacterial community analysis was conducted by extracting DNA from each sample anaerobically cultured for 5 days to determine the composition of the protected in vitro microbiota. Our taxonomic profile, along with rarefaction curves and the PCoA 3D plot, revealed that addition of OMVs extracted from the PMR$^{High}$ strain protected the bacterial community from PMB under our tested anaerobic conditions, resulting in recovery of a similar microbial composition (*Figure 7A–C*, *Figure 7—source data 1*). The total 16S rDNA genes per dried gram in the PMB-added sample (145,311 copies/dried gram) decreased by ~16% in comparison with that in the 5 days incubated control sample (173,403 copies/dried gram), while the corresponding value in the OMV-added sample was reduced by 14% (149,643 copies/dried gram) (*Figure 7A*). Surprisingly, the PMB-exposed sample showed significantly reduced numbers of 16S rDNA copies corresponding to the *Enterococcus faecium* group belonging to Gram-positive bacteria (75% lower than the control) and recovery of the *E. faecium* group occurred in the OMV-added PMB sample (*Figure 7A*). No growth inhibition with PMB was observed in both aerobically grown *Enterococcus* type strains (*E. faecium* and

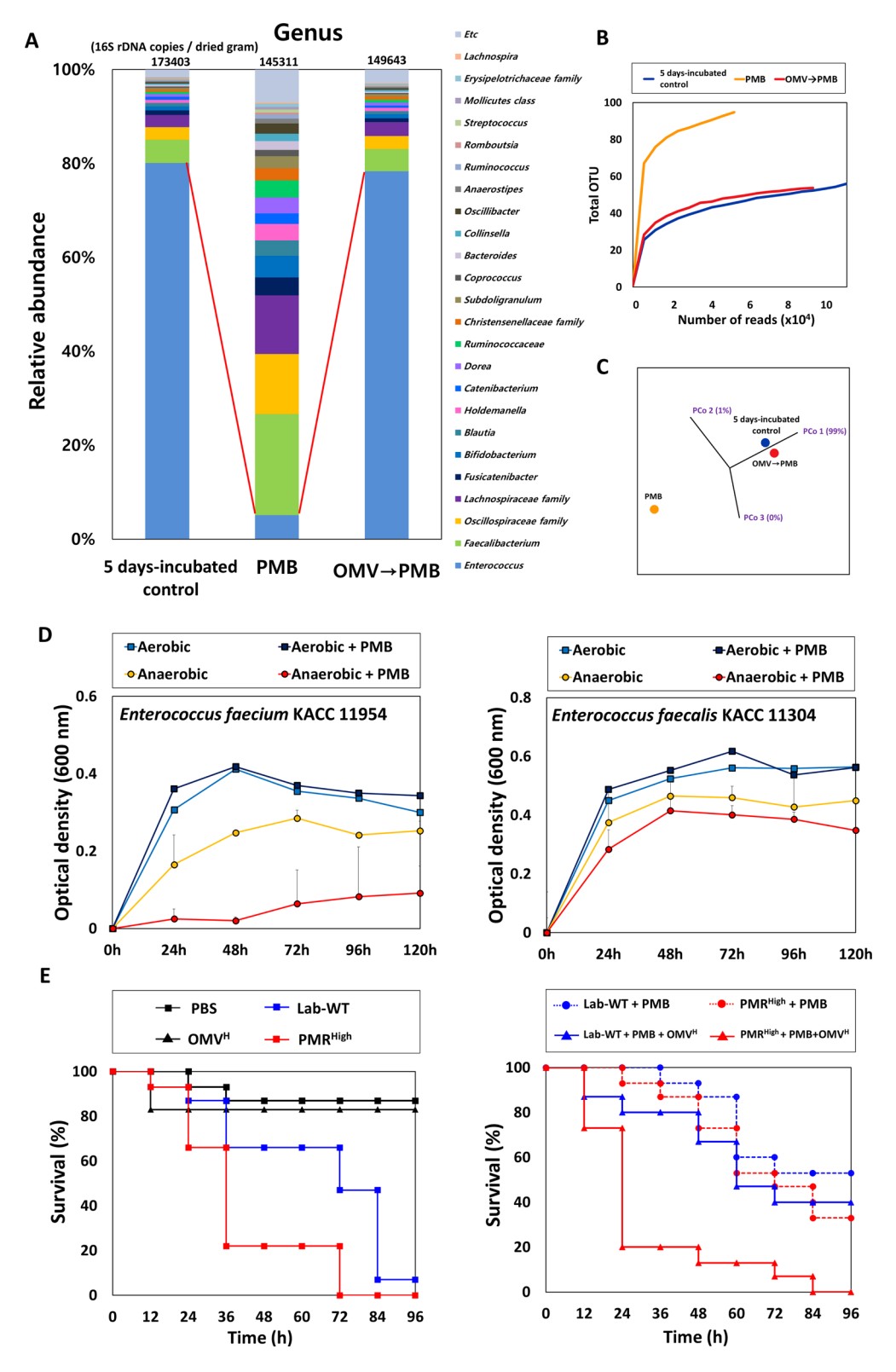

**Figure 7.** Protective effects of outer membrane vesicles (OMVs) on human fecal microbiota and in *G. mellonella* infected with *A. baumannii* strains. (**A**) The stacked bar plot shows the relative abundance of the microbiome at the genus level. The color keys indicate the 24 most abundant taxa. OMVs (25 μL/mL) were first added and allowed to stabilize, followed by incubation at 37°C after the addition of 4 μg/mL polymyxin B (PMB) for 3 hr. (**B**) The α-diversity analyses using rarefaction curves of operational taxonomic units for in vitro human fecal microbiota. (**C**) The principal coordinate analysis 3D

*Figure 7 continued on next page*

*Figure 7 continued*

plot of in vitro human fecal microbiota using log2-transformed Bray–Curtis distances. The distance between symbols on the ordination plot indicates the relative similarity in their community structures. (D) The growth curves of *Enterococcus* spp. under aerobic or anaerobic conditions. PMB treatment (4 µg/mL) was conducted daily for 5 days in identical conditions for analysis of the microbiome community (N = 3 biological replicates). (E) The *G. mellonella*-infection model was constructed to confirm the protective effects of OMV in vivo. 10 µL of OMVs, $10^6$ CFU/mL of Lab-WT or PMR^High, and 1 µg/mL PMB were mixed and injected through a syringe. For groups without OMVs, the same volume of phosphate-buffered saline was used. Mortality was defined as a lack of response or melanization for 96 hr in infected *G. mellonella* at 37°C (N = 3 biological replicates).

The online version of this article includes the following source data and figure supplement(s) for figure 7:

**Source data 1.** Protective effects of outer membrane vesicles on human fecal microbiota and growth curve of *Enterococcus* spp. under aerobic or anaerobic conditions.

**Figure supplement 1.** Measurement of community polymyxin B (PMB) minimum inhibitory concentration (MIC) in in vitro human fecal microbiota and protective effect of outer membrane vesicles (OMVs) under PMB conditions.

*Enterococcus faecalis*). Interestingly, severe growth retardation with PMB was especially observed in the anaerobic culture of *E. faecium* (*Figure 7D*, *Figure 7—source data 1*). Unlike aerobically grown cells, anaerobically grown *Enterococcus* cells might show the effects of PMB toxicity on membrane charges and composition, which warrants further examination. The survival of *G. mellonella* larvae infected with Lab-WT or PMR^High was assessed in the presence or absence of OMVs (*Figure 7E*). Larvae were infected with either Lab-WT ($10^6$ CFUs/mL) or PMR^High with or without PMB (1 µg/mL) after OMV addition. Comparison of the mortality rates demonstrated that PMR^High, which possessed higher virulence than Lab-WT, induced faster death of all tested larvae within 72 hr (*Figure 7E*). In the presence of PMB, only 10% of the *G. mellonella* larvae infected with PMR^High died within 24 hr, but 80% of the larvae died when OMVs were included (*Figure 7E*). This experiment suggested that *A. baumannii* survived longer due to the protective effect of OMVs against PMB, leading to the higher mortality of *G. mellonella*. Collectively, OMVs can protect the human pathogen *A. baumannii* from PMB under in vivo and in vitro conditions by functioning as decoys for PMB binding.

## Discussion

Production of OMVs has been linked to many important cellular behaviors, including quorum-sensing, AR, and biofilm formation in pathogenic bacteria (*Florez et al., 2017*). OMV biogenesis was thought to be also linked to the elevated hydrophobicity of the cell surface in *P. putida*, which caused greater biofilm formation (*Baumgarten et al., 2012*). OMVs are known to function as cargoes for many extracellular or intracellular materials such as virulence factors, proteins, DNA, RNA, and signaling molecules (*Bonnington and Kuehn, 2016*). Interestingly, the OMVs produced by *A. baumannii* have been shown to have a virulence factor (OmpA) and an antibiotic-resistance factor (AmpC) (*Weber et al., 2017*). OMVs are composed of PLs, OMPs, periplasmic proteins, and LPS; however, the packaging of cytosolic molecules into OMVs remains unclear (*Roier et al., 2016a*; *Jan, 2017*). Interestingly, OMVs released from CRAB could be responsible for the horizontal transfer of the carbapenem-resistance-related gene *blaNDM-1* to carbapenem-susceptible *A. baumannii* (*Chatterjee et al., 2017*). Notably, *Klebsiella pneumoniae* strains had OMs and larger OMVs harboring different lipid compositions under PMB treatment, which might contribute to the low permeability of PMB (*Jasim et al., 2018*). The following factors have been proposed to modulate OMV biogenesis: (i) diminished linkage proteins between OM and PG; (ii) increased turgor pressure due to the accumulation of cellular materials in the periplasm; (iii) modulation of the production of transmembrane proteins; and (iv) changes in membrane fluidity and rigidity (*Toyofuku et al., 2015*). The bacterial membranes of Gram-negative bacteria are also constantly modified via two-component systems, including PmrAB, in response to changes in the environment such as metals and PMB (*Tsang et al., 2017*). LptA is the periplasmic protein of the LPS transport (Lpt) complex, which bridges the IM and OM, transporting LPS across the periplasm (*Powers and Trent, 2019*). A recent study has shown that reduced LptA connections lead to overproduction of OMVs (*Falchi et al., 2018*). Decreased Lpp-PG covalent cross-linking or non-covalent OmpA-PG interactions also

enhance the production of OMVs (*Schwechheimer and Kuehn, 2015*). Disruption of the Mla complex system, which transports mislocalized OM glycerophospholipids back to the IM, results in hypervesiculation by accumulation of glycerophospholipids in the outer leaflet (*Kamischke et al., 2019*). The Bam complex is an OM complex that mediates the folding and insertion of OM proteins into the membrane (*Malinverni and Silhavy, 2011*). DolP, a lipoprotein, is known to promote proper folding of BamA, one of the essential subunits of the BAM complex, which might also play a role in OM integrity (*Ranava et al., 2021*). Deletion of *bamE* in *E. coli* or in *Salmonella enteritidis* results in a more severe defect in OM and lower OM protein levels than does loss of *bamC* (*Sklar et al., 2007*; *Fardini et al., 2009*).

Taken together, the accumulated data suggest that OMVs can be generated when decreased cross-links between the OM, PG, and IM occur in the bacterial membrane. Our omics and mutational analyses substantiated that many genes involved in the structural components of OM and IM, such as *ompA, lpp, mlaC,* and *ftsL,* play important roles in OMV biogenesis (*Figures 3* and *4*). However, it is still not clear how the DNA repair-related *udg* gene is linked to OMV production (*Figure 3*). Linkage between those deleted regions in evolved strains and OMV biogenesis remains to be investigated (*Figure 3—figure supplement 1A*). It is worth noting that antibiotic-induced genomic alterations including deletions, rearrangements, and amplification have been reported in many laboratory-evolved antibiotic-resistant bacterial strains (*Sandegren and Andersson, 2009*). Several laboratory-evolved antibiotic-resistant *E. coli* strains also appeared to carry more than 20 mutations (*Maeda et al., 2020*). Tetracycline-induced genome rearrangements led to the loss of 6.1 kb having seven full-length and two truncated genes in *E. coli* strains (*Hoeksema et al., 2018*). Norfloxacin-treated *E. coli* had ~11.5 kb deletions possibly through IS1-mediated recombination (*Long et al., 2016*). Deducing a direct link between OMV production and a single OMV-related gene might be an oversimplification because of the multifaceted control of bacterial OMV production; however, our expression analyses clearly demonstrated that the hypervesiculating PMR$^{High}$ strain showed reduced expression of many OMV-related genes, which might cumulatively affect OMV production. Several other data have also shown that bacterial OMVs contained IM proteins as well as cytoplasmic proteins (*Nagakubo et al., 2019*). The unusual OMV biogenesis recently identified in a *P. aeruginosa* model system has clearly shown that OMV production through explosive cell lysis ended up including several cytosolic components of cells (*Schwechheimer and Kuehn, 2015*). Explosive cell lysis induced by phage-derived endolysin burst the outer and IMs, leading to the presence of cytoplasmic proteins and DNA in OMVs (*Turnbull et al., 2016*). Two known mechanistic factors (PQS and MucD) for OMV production were not required for cycloserine-induced OMV formation in *P. aeruginosa*, indicating the existence of multiple pathways for OMV biogenesis (*Macdonald and Kuehn, 2013*).

OMV biogenesis in PMR$^{High}$ causes PMB resistance via the decoy activity of OMVs. OMVs have been shown to protect bacterial cells against several classes of antibiotics and physical and chemical stresses (*Kulkarni et al., 2015*; *Lee et al., 2016*). The decoy activity of OMVs was proposed in an *E. coli* model under membrane-active antibiotics such as colistin and melittin (*Kulkarni et al., 2015*; *Kim et al., 2018*). Our study also supported the role of OMVs as decoys under PMB conditions in *A. baumannii* (*Figure 5A, B*), clinical isolates (*Figure 6B*), and in the microbial community (*Figure 7A*). The in vitro human microbiota experiment clearly showed that the presence of OMVs can modulate the bacterial community under PMB conditions. The reasons underlying our unexpected observation of sensitivity of the Gram-positive *Enterococcus* strains to PMB under anaerobic conditions remain to be elucidated (*Figure 7A*). Recently, more studies suggested that polymyxins can bind to teichoic acid, which is a negatively charged compound in the OM of Gram-positive bacteria (*Brown et al., 2013*; *Rudilla et al., 2018*; *Yu et al., 2020*). We speculated that different degrees of teichoic acid-like production might occur in anaerobic culture conditions. Our *G. mellonella*-infection experiments also suggested the protective role of OMVs on *A. baumannii* cells against PMB during infection (*Figure 7C*). Likewise, *Vibrio cholerae* cells use hyperproduction of OMVs to increase their colonization and adaptation during mammalian infection (*Zingl et al., 2020*). Interestingly, the OMVs induced by the *pga* operon can stimulate the production of broad antimicrobial antibodies against *P. aeruginosa* in a mouse infection model (*Stevenson et al., 2018*). In conclusion, our data suggested that greater attention must be paid to develop new antibiotics and vaccines because the multifaceted OMV production in pathogenic *A. baumannii* strains warrants extensive investigations.

# Materials and methods

**Key resources table**

| Reagent type (species) or resource | Designation | Source or reference | Identifiers | Additional information |
|---|---|---|---|---|
| Strain, strain background (*Escherichia coli*) | DH5a | Thermo Fisher Scientific | 18265017 | |
| Strain, strain background (*Escherichia coli*) | S17-1λ *pir* | This laboratory | | |
| Strain, strain background (*Acinetobacter baumannii* ATCC 17978) | *A. baumannii* Lab-WT | This study | | |
| Strain, strain background (*Acinetobacter baumannii* Lab-WT) | *A. baumannii* PMR$^{Low}$ | This study | | |
| Strain, strain background (*Acinetobacter baumannii* Lab-WT) | *A. baumannii* (pRK-pmrB) | This study | | |
| Strain, strain background (*Acinetobacter baumannii* Lab-WT) | *A. baumannii* ΔpmrB | This study | | CRISPR-Cas9 method |
| Strain, strain background (*Acinetobacter baumannii* Lab-WT) | *A. baumannii* ΔftsL | This study | | Single-crossover recombination method |
| Strain, strain background (*Acinetobacter baumannii* Lab-WT) | *A. baumannii* Δlpp | This study | | Single-crossover recombination method |
| Strain, strain background (*Acinetobacter baumannii* Lab-WT) | *A. baumannii* ΔmlaC | This study | | CRISPR-Cas9 method |
| Strain, strain background (*Acinetobacter baumannii* Lab-WT) | *A. baumannii* ΔpgaB | This study | | CRISPR-Cas9 method |
| Strain, strain background (*Acinetobacter baumannii* Lab-WT) | *A. baumannii* Δudg | This study | | Single-crossover recombination method |
| Strain, strain background (*Acinetobacter baumannii* Lab-WT) | *A. baumannii* ΔfabH | This study | | Homologous recombination method |
| Strain, strain background (*Acinetobacter baumannii* Lab-WT) | *A. baumannii* ΔsurE | This study | | Homologous recombination method |
| Recombinant DNA reagent | pVIK112 (Plasmid) | This study | Addgene: Plasmid #V002240 | Produced by De Schamphelaire W et al. |
| Recombinant DNA reagent | pRK415 (Plasmid) | This study | Life Science Market Plasmid # PVT5709 | Produced by Keen NT et al. |
| Recombinant DNA reagent | pSGAb-km | *Wang et al., 2019* | Addgene: Plasmid #121999 | Km$^{r}$, *ColE1, WH1266* |

*Continued on next page*

*Continued*

| Reagent type (species) or resource | Designation | Source or reference | Identifiers | Additional information |
|---|---|---|---|---|
| Recombinant DNA reagent | pCasAb-apr | *Wang et al., 2019* | Addgene: Plasmid #121998 | Apr$^r$, *oriV*, broad-host-range vector |
| Commercial assay or kit | RNeasy Mini Kit | QIAGEN | 74104 | Accordance with the instructions of the manufacturer |
| Commercial assay or kit | FastDNA spin kit | MP Biomedicals | 116540600-CF | Accordance with the instructions of the manufacturer |
| Chemical compound, drug | Polymyxin B (PMB) | Sigma-Aldrich | P4932-1MU | Targets LPS |
| Chemical compound, drug | Kanamycin | Sigma-Aldrich | B5264 | Targets RNA |
| Chemical compound, drug | Apramycin | Sigma-Aldrich | A2024-1G | Targets RNA |
| Chemical compound, drug | IPTG | Gold Bio | I2481C | Induces gene expression |
| Chemical compound, drug | FM 4-64 | Thermo Fisher Scientific | T3166 | Fluoresces when bound to phospholipids |
| Chemical compound, drug | Danysl-chloride | Sigma-Aldrich | D2625-1G | |
| Chemical compound, drug | FilmTracer SYPRO Ruby | Invitrogen | F10318 | |
| Software, algorithm | CLC Genomics Workbench v.10.0.1 | QIAGEN | | |
| Software, algorithm | MUMmer | http://mummer.sourceforge.net/ | | |
| Software, algorithm | IS finder | https://isfinder.biotoul.fr/blast.php | | |
| Software, algorithm | EzGenome | http://ezgenome.ezbiocloud.net/ezg_browse | | |
| Software, algorithm | Snapgene | https://www.snapgene.com/ | | |

## Experimentally evolved stains, clinical isolates, and determination of PMB susceptibility

The bacterial strains used in this study are listed in *Supplementary file 1i*. We named *A. baumannii* ATCC 17978 wild-type strain as Lab-WT and constructed PMB-resistant strains (PMR$^{Low}$ and PMR$^{High}$) from Lab-WT. All the strains were grown at 37°C in LB broth with aeration by shaking at 220 rpm. The PMR$^{Low}$ and PMR$^{High}$ strains were constructed by transferring wild-type cells onto LB agar plates with serially higher PMB concentrations. In total, 3 *A. baumannii* experimentally evolved isolates and 40 clinical isolates were investigated. The clinical isolates were obtained from the Samsung Medical Center, Sungkyunkwan University School of Medicine, and National Culture Collection for Pathogens (NCCP). The complete genome sequence of the *A. baumannii* ATCC 17978 strain can be accessed in GenBank (accession number: CP000521, CP015122.1). Sequence reads for the experimentally evolved isolates are accessible from the NCBI sequence archives under accession numbers PRJNA530195 (Lab-WT), PRJNA530197 (PMR$^{Low}$), and PRJNA530202 (PMR$^{High}$). Nine different

classes of antibiotics, including meropenem, rifampicin, chloramphenicol, and ampicillin, were selected to test the AR. The MICs of each antibiotic were examined in all tested cells (Lab-WT, PMR$^{Low}$, PMR$^{High}$, *P. aeruginosa*, *P. putida*, *E. coli*, and *A. oleivorans*) with the broth-dilution method using 96-well plates, as previously described (*Pérez-Cruz et al., 2016*). For determination of protection tendency with OMV in clinical isolates, 40 clinical *A. baumannii* isolates were grown at 37°C in LB broth at 220 rpm until each culture reached an OD$_{600}$ of 0.8–1.0. The cell pellet was washed with phosphate-buffered saline (PBS) twice and each collected cells ($10^6$ CFU/mL) was inoculated separately into LB medium with or without 1/2 MIC of each isolate for 1 hr at 37°C. After grown cells were serially diluted to get $10^{-4}$ dilution with PBS, 10 μL of suspension spotted on an LB agar plates and incubated for 16 hr at 37°C. The protection rate (%) was calculated survived colonies (CFU) in comparison with untreated conditions of each strain.

## Quantitative reverse transcription-PCR

The expression levels of our target genes were normalized using the expression level of 16S rDNA, as previously described (*Shin et al., 2020*). The total RNAs from different *A. baumannii* strains were isolated from 5 mL of cell cultures in the mid-exponential phase (optical density at 600 nm [OD$_{600}$] of approximately 0.4) using the RNeasy Mini Kit (QIAGEN, Germany). For qRT-PCR, the cDNA was synthesized from 3 μg of RNA using the primers listed in *Supplementary file 1j*. All qRT-PCR procedures were conducted in triplicate from at least three independent cultures. The relative expression levels were compared with those of the Lab-WT strains.

## Measurement of biofilm formation

Biofilms were grown in confocal dishes (SPL Life Sciences, South Korea) at 37°C for 24 hr in LB broth. The biofilm cells were stained with FilmTracer SYPRO Ruby (Invitrogen, USA) for 30 min at room temperature (RT), protected from light, and then washed with distilled water. The obtained CLSM (Carl Zeiss, Germany) images were analyzed and modified using the Zen 2.1 (Blue edition; Carl Zeiss, Germany) software. LB broth in sterile 96-well microtiter plates (SPL Life Sciences) was inoculated in triplicate with each overnight LB-grown culture and further diluted 1:100 with LB broth. The volume of the cells was determined by converting the OD$_{600}$ value of the O/N cells. Uninoculated LB broth was used as the negative control. The microtiter plates were then incubated at 37°C for 24 hr. After removing planktonic cells, the biofilm biomass was stained with crystal violet and solubilized with 95% ethanol (v/v), after which its absorbance was measured at 595 nm. To measure OMVs-induced initial stage of biofilm formation in Lab-WT, cells were grown at 37°C in LB until an OD$_{600}$ reached around 0.6 (for 6 hr). Biofilm formation was checked after additional 4 hr incubation with PMR$^{High}$-driven OMVs (0, 12.5, or 25 μg/mL), then both crystal violet assay and CLSM observation were used.

## Lipid A extraction and analysis with MALDI-TOF mass spectrometry

*A. baumannii* strains were grown at 37°C in LB broth until each culture reached an OD$_{600}$ of 1.0. Cells were harvested via centrifugation at 7800 rpm for 15 min. Then, the cell pellet was washed with 30 mL of PBS. The supernatant was poured off and resuspended in 35 mL of a single-phase Bligh–Dyer mixture (chloroform:methanol:distilled water, 1:2:0.8; v/v). The solution was mixed by inversion and incubated at RT for >20 min to ensure complete cell lysis. Then, the mixture was centrifuged at 2000 rpm for 20 min. LPS will pellet along with proteins and nucleic acids; however, PLs, isoprenyl lipids, and small, hydrophobic peptides remain in the supernatant. The supernatant was discarded. The LPS pellet was washed with 35 mL of the single-phase Bligh–Dyer mixture and centrifuged at 2000 rpm for 20 min. The supernatant was discarded; 8 mL of mild acid hydrolysis buffer (50 mM sodium acetate, pH 4.5; 1% SDS) was added to the pellet, and it was mixed by pipetting. The solution was sonicated at a constant-duty $2\times$ cycle for a 20 s/burst, with an interval of 5 s between bursts, at 50% output. Then, the sample was boiled in a water bath for 30 min, and the bottles were removed from the water bath and allowed to cool to RT. To extract lipids after hydrolysis, the mixture was formed by adding 9 mL of chloroform and 9 mL of methanol to the SDS solution. Then, the sample was mixed by inversion and centrifuged for 10 min at 2000 rpm. The lower phase was extracted into a clean 50 mL conical tube. A second extraction was performed by adding 30 mL of the lower phase from a pre-equilibrated two-phase Bligh–Dyer mixture (chloroform:methanol:DW, 2:2:1.8; v/v) to the upper phase from the previous step. The mixture was centrifuged at 2000 rpm for 10 min. The lower

phase was extracted and pooled with the lower phase from the previous step. Then, the pooled lower phase (18 mL total) was washed by adding 34.2 mL of pre-equilibrated two-phase Bligh–Dyer upper phase to create a two-phase Bligh–Dyer mixture (chloroform:methanol:DW, 2:2:1.8; v/v). The solution was mixed and centrifuged at 2000 rpm for 10 min. The lower phase was removed to a clean glass rotary evaporator flask and the sample was dried using rotary evaporation. The lipid sample was added to 1.5 mL of chloroform:methanol (4:1, v/v) in a rotary flask, and ultrasonicated (40 s) to facilitate suspension of lipid from the sides of flask. The lipid A was dried in a nitrogen dryer and transferred to small glass sample vial. The dried sample was stored at 4°C. Lipid A was analyzed by MALDI-TOF mass spectrometry in the positive-ion mode. The matrix was a saturated solution of 2,5-dihydroxybenzoic acid (DHB). A 10 μL volume of the matrix solution was deposited on the sample plate, followed by 10 μL of the sample dissolved in chloroform:methanol (4:1, v/v).

## WGS and data analysis

WGS analyses were performed on three selective *A. baumannii* isolates using the PacBio sequencing technique (ChunLab, South Korea). The genome sequences were downloaded from EzGenome (http://ezgenome.ezbiocloud.net/ezg_browse) and were analyzed using the CLgenomics program (ChunLab). Long-read NGS sequencing techniques, such as the PacBio used in this study, may produce sequencing errors (*Rhoads and Au, 2015*). To compensate for this shortcoming, additional PCR reactions using *Pfu* DNA polymerase were performed to confirm mutational positions listed in *Supplementary file 1a* (18 genes). The primers used for PCR reactions are listed in *Supplementary file 1k*. Genomes were analyzed by using ClustalW 2.0 software, while neighbor-joining trees and amino acid compositions were constructed and calculated using MEGA6 software. The origin of replication (*oriC*) was identified using Ori-Finder (http://tubic.tju.edu.cn/Ori-Finder/). In addition to the amino acid sequence of PmrB, the LPS/lipid A biosynthesis components were compared between the isolates and the reference strain ATCC 17978. Antibiotic-resistant genes were identified using the CLC Genomics Workbench v.10.0.1 (QIAGEN). We used the Find Resistance tool (Microbial Genomics Module) with the QMI-AR, CARD, ResFinder, and PointFinder. All parameters were set as default. Insertion sequence (IS) elements were analyzed with the IS finder (https://isfinder.biotoul.fr/blast.php).

## RNA extraction, library construction, and sequencing

Both Lab-WT and PMR$^{High}$ strains were grown to exponential phase (OD$_{600}$ ~ 0.5) in LB media. For antibiotic treatment conditions, both strains were grown to the exponential phase (OD$_{600}$ ~ 0.25) in 1/2 MIC (1 μg/mL or 64 μg/mL, respectively) of PMB-supplemented LB media. Total RNA was isolated from 10 mL of cells by using the RNeasy Mini Kit (QIAGEN) according to the manufacturer's instructions. All procedures for RNA sequencing were conducted by ChunLab. The RNA was subjected to a subtractive Hyb-based rRNA removal process using the MICROBExpress Bacterial mRNA Enrichment Kit (Ambion, USA). RNA sequencing was performed with two runs of the Illumina Genome Analyzer IIx to generate single-ended 100 bp reads. Quality-filtered reads were aligned to the reference genome sequence using the CLC Genomics Workbench v.10.0.1 (QIAGEN). Mapping was based on a minimal length of 100 bp with an allowance of up to two mismatches. Relative transcript abundance was measured in RPKM. The RNA-seq data have been deposited in NCBI under Gene Expression Omnibus (GEO) accession number GSE163581.

## *A. baumannii* mutant construction

To construct Δ*ftsL*, Δ*udg*, and Δ*lpp* mutants by homologous recombination, target genes were disrupted using a single-crossover recombination method with several vectors, such as pVIK112 and pRK415, as previously described (*Shin and Park, 2015*). The primers used in this study are listed in *Supplementary file 1j*. In general, *Eco*RI and *Kpn*I restriction enzymes were used for all the knockout and overexpression mutants. Δ*pmrB* is a *pmrB* mutant of Lab-WT strains. To construct the mutants, fragments from PCR amplification and gel extraction were inserted into the pVIK112 vector via ligation as the first step. The pVIK112-fragment unified vectors were then transformed into *E. coli* S17-1λ pir. Finally, the amplified plasmids were inserted into the Lab-WT and PMR$^{High}$ strains of *A. baumannii* ATCC 17978 by electroporation. To ensure that homologous recombination had occurred in *A. baumannii* ATCC 17978, PCR verification was conducted using the *pmrB* OE-F/MCS-R primer

pairs. The MCS-R primer was designed based on the sequence of the pVIK112 plasmid. To construct Δ*pmrB*, Δ*mlaC*, and Δ*pgaB* by the CRISPR-Cas9 method, the pCasAb-apr (cat. no 121998) and pSGAb-km (cat. no 121999) plasmids were obtained from Addgene (*Wang et al., 2019*). A pair of 20 bp spacer oligos and 80-nt oligos were designed to target the genomic locus and donor repair template for editing. A designed 20 bp spacer was phosphorylated and annealed to the pSGAb-km plasmid with a Golden Gate assembly reaction. Then, the plasmids were transformed into *E. coli* DH5a-competent cells, followed by plating onto an LB agar plate containing 100 µg/mL, and incubated at 37°C for 16 hr. Successful cloning of the spacer was verified by PCR with the primers of Spacer-F/M13R and by sequencing with the primer of M13R (5′-CAGGAAACAGCTATGACC-3′). Then, 1 M of IPTG was added into *A. baumannii* Lab-WT harboring pCasAb-apr to induce the expression of the RecAb recombination system and Cas9 nuclease. After incubating at 37°C for 2 hr, the same method was used to prepare the competent cells as described previously (*Wang et al., 2019*). Then, 200 ng of the spacer-introduced pSGAb-km plasmid and 300 µM ssDNA (donor repair template) were co-transformed to the IPTG-induced *A. baumannii* Lab-WT cells harboring the pCasAb-apr plasmid by electroporation. The cells were plated onto an LB agar plate containing 100 µg/mL apramycin and 50 µg/mL kanamycin, and the plate was incubated at 37°C overnight. Successful editing was verified by PCR and sequencing. For plasmid curing, a colony was plated onto an LB agar plate containing 5% sucrose at 37°C overnight. Then, the colony were streaked onto LB agar plates without the antibiotics to confirm curing. A detailed genome-editing protocol has been provided in the file (*Figure 4—figure supplement 1*).

## Transmission electron microscopy

TEM was conducted at the Korea University Medical Research Center (https://medicine.korea.ac.kr/web/msrc/home). The experimental procedure was in accordance with the instructions of the manufacturer. Briefly, for bacterial cell imaging, cells were grown for 18 hr in LB media. Aliquots were transferred onto carbon-coated copper grids (Formvar; Ted PELLA, Canada) and fixed with 2.5% glutaraldehyde diluted in 50 mM sodium cacodylate (pH 7.2). After washing three times with 3% saccharose, the cells were observed. For OMV imaging, aliquots of purified OMV samples were dispensed on carbon-coated copper grids. Excess liquid was discarded from the grids and the samples were negatively stained with phosphotungstic acid (3% [wt/vol]) for 5 min and then dried with filter paper. The specimens were examined using an H-7100 microscope (Hitachi, Tokyo, Japan) operating at an accelerating voltage of 75 kV and at magnifications of 40,000× for cells and 30,000× for OMVs.

## OMV quantification

The lipid content of the OMV samples was indirectly quantified using the lipophilic dye FM4-64 (*N*-(3-triethylammoniumpropyl)−4-(4-(dibutylamino) styryl) pyridinium dibromide; Molecular Probes, Eugene, OR, USA, and Life Technologies, Carlsbad, CA, USA). The lipophilic dye FM4-64, which intercalates into the OM, is commonly used to stained and observed membrane of bacteria (*Toyofuku et al., 2017*; *Rojas et al., 2018*). To measure OMVs using this dye, cells were incubated for 16 hr at 37°C and 220 rpm in 5 mL of LB. The cultured cells were harvested, and the supernatant was filtered through a 0.22 µm syringe filter. Aliquots of the filtrate (100 µL) containing OMVs were transferred to the wells of a 96-well dark plate. The FM4-64 dye was added to each well at a final concentration of 1 µg/mL. After excitation at 510 nm, the fluorescence emission at 610 nm was measured using a fluorescence spectrophotometer (TECAN, Männedorf, Switzerland) using 5 nm slit widths for both excitation and emission. To quantify the protein contents of OMVs, all the samples were standardized using the Bradford assay (Bio-Rad Laboratories, Protein Assay Dye Reagent), and triplicates were used. For the DNA contents of OMVs, the NanoPhotometer (Implen, Germany) was used to quantify the concentration of DNA. To quantify the LPS content of OMVs, purpald assays using the 2-keto-3-deoxyoctonate ammonium salt (Kdo) standard (Sigma-Aldrich) were performed as previously described (*Lee and Tsai, 1999*).

## Raman spectroscopy analysis

Isolates were grown in LB broth for the exponential phase at 37°C, and the cell pellet was washed with PBS twice. Each bacterial suspension was dropped into an aluminum dish (width, 1.75 in; depth,

0.38 in) for spectral collection. After drying the plate for 2 hr, each spot was washed with 10 µL of PBS. Raman spectra of the isolates were collected using a confocal Raman imaging system (Xper-Ram35V; Nanobase) assembled with 3-port excitation 532 nm DPSS laser (LTL-532RL; Leading Tech), microscope body (Olympus BX43; Olympus), spectrometer (XPE-35 VPHG; Nanobase), and charge-coupled device (Atik 428EX; Atik). Each spectrum was the sum of 25 s of acquisition time with 2 mW of laser power and collected for 15–20 single cells of the *A. baumannii* isolate.

## Isolation of pure OMVs and flow cytometric analysis for detecting OMVs

The procedure used to isolate the OMVs is presented in *Figure 5—figure supplement 1*. OMVs were isolated from PMR$^{High}$ bacteria. The isolation was confirmed using FM4-64 staining and flow cytometric analysis. To extract OMVs from PMR$^{High}$ strains, 600 mL of cells were grown for 16 hr to obtain a robust and quantifiable amount of OMVs. Overnight bacterial culture was grown to an OD$_{600}$ of 0.8–1.2 and the culture was centrifuged at 7800 rpm at 4°C for 30 min. The supernatant of each culture was sequentially filtered through 5 µm hydrophilic polyvinylidene difluoride and 0.45 µm Millex membrane filters (Millipore, Billerica, MA, USA). The resulting filtrates were concentrated with centrifugation (7800 × $g$) at 4°C for 30 min using a 10 kDa molecular weight cutoff Amicon Ultra-15 centrifugal filter unit (Millipore). Each concentrated filtrate was centrifuged (100,000 × $g$, 4°C, 3 hr) in a tabletop Optima ultracentrifuge (Beckman Coulter). Pelleted OMVs were suspended in 100 µL of PBS. The OMVs were quantified by using Bradford assay. The confirmed OMV pellet was spread on LB agar plates to confirm the cell-free status. The OMVs were counted by using a flow cytometer (BD Accuri C6 Plus). The forward scatter threshold value was 10,000. Three independent experiments were conducted for each condition, with 50,000 cells typically analyzed per experiment.

## Synthesis of dansyl-PMB and measurement of PMB binding affinity by CLSM

PMB sulfate (40 mg, Sigma-Aldrich) was dissolved in 1.2 mL of 0.1M NaHCO$_3$, and dansyl-chloride (10 mg, Sigma-Aldrich) was dissolved in 0.8 mL of acetone. Dansyl-chloride was added to PMB and placed in the dark for 90 min at RT. After incubation, the mixture was loaded onto a Sephadex G-50 column (50 × 2.5 cm) equilibrated with 10 mM Na-phosphate buffer (pH 7.1) containing 0.145M NaCl, and 5–6 mL fractions from the column were assessed. The dansyl-PMB appears as a broad peak ahead of the unreacted dansyl-chloride peak. The location of the dansyl-PMB in the collected fractions was determined by holding a UV lamp over the fractions and looking for fluorescence. The fluorescence of the dansyl-PMB was yellowish, while the unreacted dansyl-chloride showed more blue-green fluorescence. The fractions containing dansyl-PMB were extracted into approximately 1/2 vol of *n*-butanol. The butanol was then evaporated to dryness in a glass Petri dish placed inside a desiccator, which was then evacuated and placed at 37°C for 24 hr. The dried dansyl-PMB was dissolved in 3 mL of buffer (5 mM HEPES, pH 7.0) and stored at −20°C. CLSM was performed using LSM 770 (Carl Zeiss Microscope, Jena, Germany). The exponential phase of the 5 mL bacterial culture was washed and resuspended with PBS. The cells were stained with dansyl-PMB (2 or 4 µg/mL) for 30 min at 37°C.

## Protective effect of OMVs in the presence of PMB

To confirm the protective effect of OMVs on LB agar plates, cultured cells were inoculated on media containing 0, 0.5, 1, 2, or 4 µg/mL PMB in the absence or presence of OMVs from PMR$^{High}$. Each culture was serially diluted 1:10 and inoculated beginning in the first spot. In the liquid medium, the protective effect of OMVs against PMB was monitored by assessing the growth curves of OMV-treated Lab-WT, PMR$^{Low}$, and PMR$^{High}$ cells and clinical isolates (designated F-1025, F-1208, F-1379, F-1410, and F-1629) from patients treated at Sungkyunkwan University Hospital. PMB was used at concentrations of 0, 1, 2, and 4 µg/mL. The cultured cells (10$^6$ CFU/ml) were inoculated separately into LB medium containing different concentrations of PMB (0, 1, 2, and 4 µg/mL) and PMR$^{High}$-derived OMVs (25 µL/mL). Bacterial growth was monitored at OD$_{595}$ at 2 hr intervals for 72 hr. Experiments were performed using bacterial cultures from three independent batches. The bacterial cultures were shaken for 10 s inside the instrument (TECAN) every 10,000 s.

## Analysis of fecal microbiota

The fecal sample was obtained from a woman aged 25 years, collected in sterilized plastic tubes, and suspended at a dilution of 1/10 in PBS (pH 7.4). The fecal solution (5 mL) was added to an anaerobic bottle, injected with 80% $N_2$ + 20% $CO_2$ gas, and sterilized. Subsequently, fecal microbiota (50 μL, 1/100 of the fecal solution) were inoculated and cultured at 37°C. Analysis of bacterial communities revealed whether the protected bacteria depended on the type of OMV in the bacterial community (*Figure 6A*). PMB (4 μg/mL) and OMVs (25 μL/mL) of equal concentrations were administered daily into the fecal samples suspended in PBS; the sample was stabilized for 3 hr by applying OMVs first, and then treated with PMB and incubated at 37°C for 5 days. Then, total community DNA was extracted using a FastDNA spin kit for soil (MP Biomedicals, USA), and the DNA yield was quantified using a NanoDrop spectrophotometer (BioTek, USA). The V3-V4 hypervariable region of 16S rRNA gene from the genomic DNA was amplified using the primers 341F (5′-TCGTCGGCAGCGTC-AGA TGTGTATAAGAGACAG-CCTACGGGNGGCWGCAG-3′) and 805R (5′-GTCTCGTGGGCTCGG-AGA TGTGTATAAGAGACAG-GACTACHVGGGTATCTAATCC-3′). The amplified products were confirmed by agarose gel electrophoresis. The amplicons were purified by Agencourt AMPure XP (Beckman Coulter, Republic of Korea) and quantified using a Quanti-iT Picogreen dsDNA Assay kit. Equimolar concentrations of each amplicon from the different samples were pooled and purified using Agencourt AMPure XP (Beckman Coulter). All sequencing procedures were conducted by ChunLab. The sequences obtained were compared and classified using the EzTaxon Database (http://www.ezbiocloud.net). The operational taxonomic units (OTUs) among the samples were obtained with a taxonomic composition using the CLcommunity program (ChunLab). The bacterial community data have been deposited in NCBI under Sequence Read Archive (SRA) accession numbers SRX9819399 PRJNA689940, SRX9819397 PRJNA689944, and SRX9819398 PRJNA689944.

## *G. mellonella*-infection model

*G. mellonella* larvae were obtained from SWORM (Cheonan, Republic of Korea). Healthy *G. mellonella* larvae weighing 200 mg were starved for 4 hr at 20°C before injection. Then, the larvae were placed on ice and injected via the last left proleg with 10 μL of OMV solution (25 μL/mL) using 31-gauge, 6-mm-long needles (BD Ultra-Fine insulin syringes). Infected larvae were divided into the following four experimental groups with 15 larvae per group: (i) inactive control groups, which received 10 μL of PBS or OMVs; (ii) bacterial treatment groups, which received either $1 \times 10^6$ CFU/larvae of *A. baumannii* ATCC 17978 Lab-WT or PMR$^{High}$; (iii) bacteria and antibiotic treatment groups, which received $1 \times 10^6$ CFU/larvae of Lab-WT or PMR$^{High}$ and 1 μg/mL PMB; and the (iv) OMV treatment group, which received $1 \times 10^6$ CFU/larvae of Lab-WT or PMR$^{High}$, 1 μg/mL PMB, and OMVs. PBS was added to each group up to 10 μL. The larvae were incubated in a Petri dish (90 × 15 mm, SPL Life Sciences) with 100 mg of wheat bran powder (MG Natural, Republic of Korea) at 37°C in air for 96 hr and inspected and scored every 12 hr for death, failure to move in response to touch, and melanization.

## Acknowledgements

We thank the group of Dr. Kwansoo Ko for providing the 5 clinical isolates from patients at Samsung Medical Center, Sungkyukwan University School of Medicine, and 35 clinical isolates from the National Culture Collection for Pathogens (NCCP). *Funding:* This work was supported by grants from the National Research Foundation of Korea (no. NRF-2020M3A9H5104237). The funders had no role in study design, data collection or interpretation, or the decision to submit the work for publication.

# Additional information

## Funding

| Funder | Grant reference number | Author |
| --- | --- | --- |
| National Research Foundation of Korea | NRF-2020M3A9H5104237 | Woojun Park |

The funders had no role in study design, data collection and interpretation, or the decision to submit the work for publication.

## Author contributions
Jaeeun Park, Misung Kim, Conceptualization, Investigation, Methodology, Writing - original draft, Writing - review and editing; Bora Shin, Conceptualization, Investigation, Methodology, Writing - original draft; Mingyeong Kang, Jihye Yang, Investigation, Methodology; Tae Kwon Lee, Methodology; Woojun Park, Conceptualization, Supervision, Funding acquisition, Writing - original draft, Project administration, Writing - review and editing

## Author ORCIDs
Tae Kwon Lee  http://orcid.org/0000-0003-3845-7316
Woojun Park  https://orcid.org/0000-0002-3166-1528

## Decision letter and Author response
Decision letter https://doi.org/10.7554/eLife.66988.sa1
Author response https://doi.org/10.7554/eLife.66988.sa2

## Additional files

### Supplementary files
• Supplementary file 1. Tables listing genomic and trasncriptomic data, antibiotic resistance of all tested isolates, and oligonucleotide sequences used in this study. (**a**) Findings from sequence analysis of mutated genes in the genomes of PMR$^{Low}$ and PMR$^{High}$ isolates. Mutations were identified using the MUMmer program. Additional PCR reactions using *Pfu* DNA polymerase were performed to confirm mutational positions. All position numbers are based on the genes of *A. baumannii* ATCC 17978. (**b**) Whole-genome sequence analyses of Lab-WT, PMR$^{Low}$, and PMR$^{High}$ isolates using the PacBio sequencing technique. The genome sequences were downloaded from EzGenome and then their chromosomes were analyzed using the CLgenomics program. (**c**) Lists of deleted regions in the chromosomes of PMR$^{Low}$ and PMR$^{High}$. The genome sequences were compared to the reference genome, *A. baumannii* ATCC 17978, and whole genomes were aligned using Snapgene. (**d**) Identification of antibiotic-resistance genes (ARGs) in Lab-WT, PMR$^{Low}$, and PMR$^{High}$ strains. Totally 20 ARGs were detected using the CLC Genomics Workbench v.10.0.1 (QIAGEN, Germany). (**e**) Identification of insertion sequence (IS) elements in the Lab-WT, PMR$^{Low}$, and PMR$^{High}$. All parameters were set as default. IS elements were analyzed using the IS finder (https://isfinder.biotoul.fr/blast.php). (**f**) All information on the acquisition of RNA-seq data in this study. Total RNAs were extracted from exponentially grown Lab-WT, Lab-WT strain + polymyxin B (PMB), PMR$^{High}$ and PMR$^{High}$ strain + PMB. RPKM represents reads per kilobase of transcript per million mapped sequence reads. (**g**) List of genes that were up- or downregulated in PMR$^{High}$, Lab-WT + PMB, and PMR$^{High}$ + PMB in comparison with control strains. Up- (fold change > 1.4, RPKM > 50) or downregulated (fold change < 0.7, RPKM > 50) genes were indicated. The intersection area and numbers are described in *Figure 3—figure supplement 1*. All locus tags are based on the genes of *A. baumannii* ATCC 17978. *TM = transmembrane region, ND = not detected. (**h**) Relative abundance of proteins in three conditions that increased outer membrane vesicles (OMVs): Lab-WT + PMB, PMR$^{High}$ + PMB, and PMR$^{High}$ strains compared to Lab-WT and PMR$^{High}$. The quantity of protein in each spot was normalized to the total valid spot intensity. By comparing each gel image of proteins in strains under overproduction of OMVs, significantly changed spots were selected based on up- (fold change > 2, intensity > 1000) or down-expressed (fold change < 0.5, intensity > 1000) protein. The total number of final selected proteins for MALDI-TOF analysis was 43 spots. *ND = not detected. (**i**) Minimum inhibitory concentrations (MICs) of antibiotics used in this study. (**j**) Bacterial strains, plasmids, and primers used in this study. (**k**) Primers used in this study. Additional PCR reactions using *Pfu* DNA polymerase were performed for confirming mutational positions listed in (**a**) (18 genes).

• Transparent reporting form

## Data availability

1. Sequence reads for the experimentally evolved isolates are accessible from the NCBI sequence archives under accession numbers PRJNA530195 (Lab-WT), PRJNA530197 (PMRLow), and PRJNA530202 (PMRHigh). 2. The RNA-seq data have been deposited in NCBI under Gene Expression Omnibus (GEO) accession number GSE163581. 3. The bacterial community data have been deposited in NCBI under Sequence Read Archive (SRA) accession number SRX9819399 PRJNA689940, SRX9819397 PRJNA689944 and SRX9819398 PRJNA689944.

The following datasets were generated:

| Author(s) | Year | Dataset title | Dataset URL | Database and Identifier |
|---|---|---|---|---|
| Shin B | 2018 | Acinetobacter baumannii strain:Lab-WT Genome sequencing | https://www.ncbi.nlm.nih.gov/bioproject/530195 | NCBI BioProject, PRJNA530195 |
| Shin B | 2018 | Acinetobacter baumannii strain:PMR-Low Genome sequencing | https://www.ncbi.nlm.nih.gov/bioproject/PRJNA530197 | NCBI BioProject, PRJNA530197 |
| Shin B | 2018 | Acinetobacter baumannii strain: PMR-High Genome sequencing | https://www.ncbi.nlm.nih.gov/bioproject/530202 | NCBI BioProject, PRJNA530202 |
| Park J | 2020 | Transcriptome of Acinetobacter baumannii with or without polymyxin B | https://www.ncbi.nlm.nih.gov/geo/query/acc.cgi?acc=GSE163581 | NCBI Gene Expression Omnibus, GSE163581 |
| Kim M | 2021 | Fecal microbiota control | https://www.ncbi.nlm.nih.gov/sra/?term=SRX9819399+PRJNA689940 | NCBI Sequence Read Archive, SRX9819399+PRJNA689940 |
| Kim M | 2021 | PMB | https://www.ncbi.nlm.nih.gov/sra/?term=SRX9819397+PRJNA689944 | NCBI Sequence Read Archive, SRX9819397+PRJNA689944 |
| Kim M | 2021 | OMV->PMB. | https://www.ncbi.nlm.nih.gov/sra/?term=SRX9819398+PRJNA689944 | NCBI Sequence Read Archive, SRX9819398+PRJNA689944 |

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
