## [Decision Letter]

**Acceptance summary:**

This study looks at the role of outer membrane vesicles produced by the major human pathogen *Acinetobacter baumannii* in the resistance to the antibiotic polymyxin. By doing extensive omics and mutational studies, the authors find that these vesicles can bind polymyxin B and protect both producing cells and entire bacterial communities from the bactericidal effect of this antibiotic.

**Decision letter after peer review:**

[Editors’ note: the authors submitted for reconsideration following the decision after peer review. What follows is the decision letter after the first round of review.]

Thank you for submitting your work entitled "A novel decoy strategy of *Acinetobacter baumannii* for polymyxin resistance" for consideration by *eLife*. Your article has been reviewed by 3 peer reviewers, including María Mercedes Zambrano as the Reviewing Editor and Reviewer #1, and the evaluation has been overseen by a Senior Editor. The following individual involved in review of your submission has agreed to reveal their identity: John Boyce (Reviewer #3).

Our decision has been reached after consultation between the reviewers. Based on these discussions and the individual reviews below, we regret to inform you that your work will not be considered further for publication in *eLife*.

We found that while the work is both novel and very relevant to the field, there were several concerns regarding the interpretation of results and the subsequent claims that make this not suitable for publication. Some of the major issues involve the way in which data was collected and presented (such as MICs, evidence for OMVs, experiments with stool samples, and statistics). Most relevant, however, is the lack of convincing evidence for the role of OMVs in polymyxin resistance, such as polymyxin binding or changes in the LPS, to support the claim for OMV protection in this species.

*Reviewer #1:*

The authors explore the role of *A. baumannii* outer membrane vesicles (OMV) in resistance to polymyxin B (PMB). They obtain a laboratory-derived resistant strain that has altered expression of genes involved in OMV biogenesis and produces more OMVs that show protection against polymyxin B both in vitro and in vivo. This includes a MIC increase for several clinical strains and enhanced survival of *A. baumannii* in a Galleria mellonella assay. The results are consistent with the role of OMVs in other bacteria, contribute additional information regarding protection of *A. baumannii* in different experimental conditions, and suggest that OMVs protect by acting as decoys for PMB binding, although the precise mechanisms remain unclear.

1. The results are interesting, but I am concerned with the specificity of the observations. In the absence of experiments with OMVs from other strains, it is difficult to assess if the results are due to this strain or if OMVs from other strains could also work (such as OMVs from the low and WT strains, the latter used also in the fecal experiment). This is also important in terms of the mechanisms involved. Given the absence of changes in surface charge (zeta potential) when cells are exposed to PMB (Figure 1C, D), what do the authors suggest is the main reason for the observed protective effect on *A. baumannii* cells?

2. The work with the feline microbiota is confusing and raises many questions, partly because of the complexity of dealing with mixed communities. The results indicate that certain taxa/groups (as identified by peaks) change when PMB / OMVs are administered. Some peaks seem to increase upon treatment with OMVH, while at least one peak increased when treating with OMVW, yet this is not mentioned in the text. How can this be explained? Are there no Gram-negative bacteria in this fecal sample and if so, are these affected or not by PMB/OMVs? While these experiments could suggest that these OMVs affect various cell types, couldn't addition of any antibiotic and any OMVs also trigger changes? It is therefore unclear if the changes, assumed as protection, are specific to these OMVs.

3. The authors mention binding to PMB as a strategy (eg, Lines 39-40 and Line 278), even though there is no direct evidence provided. It would be best to tone down these claims. It is also unclear how the work leads to the conclusion regarding novel antibiotics and "multifaceted" OMV production (Line 338). Please revise the sentence to more accurately reflect the findings of this study.

*Reviewer #2:*

In the article submitted to *eLife* by Shin et al., the authors investigate mechanisms of *Acinetobacter baumannii* resistance to polymyxin B. They isolate random mutations that occur on the genome as a result if culturing A.b. in media with PMB. They find isolates that have low and high levels of PMB resistance. Whole genome sequencing identifies nearly 20 non-synonomous mutations in different genes. Substitutions in pmrB were common between two isolates. They generate deletion mutants and overexpression genotypes for PmrB, and also contruct deletions in the other genes. The authors move on to characterize PmrB-dependent production of OMVs. The reason for focusing on this gene in favor of the others is not firmly established. The data that follow are not conducted with a sufficient level of rigor to award acceptance to this journal. No statistical analysis is presented. Significance of the data is absent, making the impact of the study very minimal.

To my knowledge, the work demonstrating the involvement of PmrB in OMV formation for this important pathogen is novel and consistent with recent data that suggest modified LOS molecules might promote or prevent vesicle formation. In general the work is well written and easy to follow conceptually. However, the level of rigor applied to this study falls well short of the requirements for this journal. In its current condition, this article is not of sufficient impact to warrant publication in this journal.

Data regarding PMB resistance is presented as MICs and growth curve measurements. No bacteriacidal assay is used, which is a major weakness and oversight. The authors report no cfu data for the bacteria that have been treated. The data is over interpreted.

Measuring effects of PMB on bacterial growth on agar media is technically problematic due to the contact dependent killing activity of PMB.

No statistics are provided to support significance of the findings.

The authors do not use appropriate assays to assess whether the particles they are measuring are actual OMVs. Western blots and enzymatic assays for particular proteins that localize or do not localize to the OMV are absent and must be included.

The authors offer no explanation for why the PMB resistant mutant also acquire multidrug resistance (rif, cat, amp etc.). This line of interrogation is abruptly ended. Do the other mutations contribute to this mechanism?

The model does absolutely nothing for the paper as it does not even speculate as to how a PmrB modified envelope would cause vesicle formation.

The experiment where *A. baumannii* OMVs are added to stool samples of a cat is not rationalized sufficiently or conducted in a manner that one could specifically attribute the effects to the OMVs. A more appropriate study would involve a mouse model of disease. *A. baumannii* does not colonize the intestine so the relevance of this model is minimal.

Fm464 staining to confirm OMV isolation is insufficient to establish that these are actual OMVs. Blotting for OmpA an OM protein and some IM protein would provide stronger support.

Similarly quantifying the lipid content with FM4-64 is not rigorous as the dye is non-specific.

*Reviewer #3:*

This paper describes an analysis of the role of outer membrane vesicles (OMVs) in polymyxin resistance in *Acinetobacter baumannii*. Firstly, polymyxin-resistant *A. baumannii* strains were selected by growth on increasing concentrations of polymyxin and then genome sequencing showed the presence of a number of mutations. Both resistant strains had two pmrB mutations known to increase PmrC expression and give PEtn addition to LPS. The strains, especially PMRHigh, showed increased OMV production and addition of purified OMVs from this strain could increase polymyxin resistance of other sensitive strains. This is a very interesting paper worthy of further investigation. However, I feel that this paper falls short of showing a convincing role for OMVs in polymyxin resistance in this species. Without further support of presence/absence of PEtn on resistant membranes and OMVs released by the PMRHigh strain, the idea that OMVs are protective seems counterintuitive; if the LPS from this PMRHigh strain has PEtn addition (highly indicated by mutations and PmrC increased expression) then the released OMVs should not bind polymyxin due to the reduced charge. Indeed, it might be expected that OMVs from WT would be more protective as they would bind polymyxin.

1. Currently, the whole genome sequencing analysis is unconvincing. Firstly, the genome sizes and number of ORFs are quite different between parent and derivative resistant strains. This seems very unusual and is likely due to different success in de novo assembly. It would have been good to use reference assembly against the parent strain as the derivatives should be almost identical. Given the likely problems with assembly, at least a number of the proposed mutations should be checked by PCR and Sanger sequencing.

2. The two common mutations in PmrB, and the increased expression of PmrC, indicate that polymyxin resistance is likely to be primarily due to PEtn addition to LPS and concomitant reduction in neg charge and polymyxin binding. However, this needs to be shown by LPS MS. Indeed, I think it is essential to look at the LPS structure in both the parent sensitive cells, resistant derivatives and the OMVs from each strain.

3. Mutation of pmrB in the PMRHigh strain failed to return the strain to fully sensitive, which does argue in favor of PEtn addition not being the primary reason for resistance. However, this would be a clearer experiment if the actual transferase gene (pmrC) was inactivated.

4. Line 161-163. Wouldn't calculation of surface charge in the presence of polymyxin be confounded by cell death in sensitive strains?

5. Line 163-165. The fact that both resistant strains were not resistant to other antibiotics suggests in favor of PEtn addition as this is a polymyxin-specific mechanism. Why is this listed as "interesting".

6. What is the proposed explanation for the high number of released OMVs in the ftsL mutant but no increase in polymyxin resistance?

7. Line 22 suggests that, "as expected", mutations in genes associated with cell wall cross linking such as lpp showed reduced OMV secretion. Isn't this at odds with line 106 which say that reduction in Lpp production disrupts the cross-linkage of PG on the membrane, thereby significantly contributing to the formation of OMVs.

8. For conclusive assessment of the role of OMVs on altered microbiota, full metagenomic 16s rRNA should be conducted, rather than terminal restriction fragment length polymorphism analysis, which does not capture enough diversity.

9. For Galleria mortality experiments, in my opinion bacterial counts should be determined

10. In my opinion, for conclusive understanding of the role of OMVs in decoy resistance, polymyxin binding should be assessed.

[Editors’ note: further revisions were suggested prior to acceptance, as described below.]

Thank you for submitting your article "A novel decoy strategy for polymyxin resistance in Acinetobacter baumannii" for consideration by *eLife*. Your article has been reviewed by 3 peer reviewers, and the evaluation has been overseen by a Reviewing Editor and Gisela Storz as the Senior Editor. The reviewers have opted to remain anonymous.

Essential Revisions:

The manuscript has improved with the inclusion of new data, but the authors should include more detailed information and take care with the interpretation of some of their results.

1. Important discussion of other studies is missing, e.g., Jasim et al. about OMVs and polymyxins in *Klebsiella* (PMID: 30103446), or Macdonald et al. about OMVs in *Pseudomonas* (PMID: 23625841), etc.

Please include statistical information required to correctly interpret the results.

2. Line 152: it should be noted that the observed 1.2-fold higher pmrC expression in PMRLow was not statistically significant. Indeed, 1.2-fold increase is below the usual cut-off values used for transcriptomics.

3. Most of the reported transcriptomics data (e.g. Figure 3B) has no statistics associated with it, so it is not possible to know whether observed differences are significant. Add false discovery rate values or CVs to the fold-change data.

4. The number of replicates should be noted in the figure legend for all experiments. E.g. for Figure 4. it seems unlikely that there is a statistically significant difference between protein Bradford assay for LabWT and deltapmrB unless there are a large number of replicates here.

5. The addition of Raman spectroscopy and the specificity of its interpretation are unconvincing.

Several issues were brought up regarding the transcriptomics (in addition to statistics) and genomics data that require further clarification.

6. There is no clear explanation for why different SNVs were observed in the transcriptomic data compared to the genomic data. Were these mutations found in a high percentage of the mapped reads?

7. The rationale for the RNA-Seq experiments is unclear and there were only a few genes dysregulated. Therefore, the value of this experiment is unclear.

8. The mutations in mRNA are troublesome as e.g., a master regulator of stringent stress response and cell division were affected and it is not clear how this is further linked to OMVs and polymyxins. Many mutations (including master regulators) in the polymyxin-resistant mutant strain could affect the observed phenotypes.

9. Why do WT and PMR-resistant strains have different genome sizes? A difference of >28 kb and 11 genes sound a lot – what was deleted in these strains specifically? There are also a lot of other important mutations in the PMR high strain that could explain the growth phenotype and this makes the whole data interpretation regarding OMVs questionable.

10. The CRISPR-Cas9 method needs to be fully detailed in Materials and methods and it should be noted which mutants are from homologous recombination and which from CRISPR.

Biofilm experiments and interpretations should be further clarified. Please address the following regarding biofilm experiments:

11. The biofilm data is not convincing and should be quantified for multiple areas. There is no evidence that OMVs are associated with biofilms in this study.

12. Supplemental Figure 1. Was the increase in biofilm seen due to excess EPS formation or an increase in viable bacteria? Was any sonication done to see if the actual number of bacteria at the timepoint chosen increase in the evolved mutant with PMB?

13. There are other mechanisms of colistin (polymyxin) resistance seen including transmissible elements such as MCR-1 which encodes a phosphoethanolamine transferase leading to increased positive charge on the membrane. Is it known whether other mechanisms of resistance would also lead to increased OMV formation? If not, it might be good to mention in the discussion about whether the mechanism underlying increased OMV production would be applicable to other forms of polymyxin resistance.

14. Did the authors determine whether OMVs from other pathogens would result in a similar decoy effect or would this be expected to be specific to polymyxin stimulated OMVs in Acinetobacter?

15. It is not clear why the authors mention DNA relaxation, but do not really explain how this is linked to whole story.

16. The document includes a large amount of experimental data that makes it hard to follow and could therefore be improved for clarity. In addition, some of the experimental information is not reflected in the abstract. Finally, authors should take care to avoid speculations.

– For example, the Results section starts with a speculation about the lower growth rate, which is inappropriate but also makes the reader wonder whether other mutations could be the cause of the growth defect. It becomes clearer reading further when the WGS is described.

– There is further speculation in the results that should not be there and lots of the results/discussion feel out of context.

Please clarify some of the figures presented.

17. Figure 1. Panel D TEM is not particularly quantitative. It is difficult to claim that there a larger number of OMVs in the PMR(high) strain. Was this quantitated visually in some way? If not, it might be helpful to include supplemental TEM images at a lower magnification to help support this claim.

18. Figure 3B (expression vs genomic position image) should be relative to log base 2 fold-change so that up- and down-regulated genes are on a similar scale.

19. It would be useful to have a merged image of Figure 5D.

20. Figure 6 Panel A. This panel is a bit confusing. I am not sure what the number scale at the top of the graph represents versus the number scale at the bottom of the panel. The dots appear blue but the authors mention black dots? I assume these are the dots they are alluding to but this needs to be clarified.

---

## [Author Response]

[Editors’ note: the authors resubmitted a revised version of the paper for consideration. What follows is the authors’ response to the first round of review.]

We found that while the work is both novel and very relevant to the field, there were several concerns regarding the interpretation of results and the subsequent claims that make this not suitable for publication. Some of the major issues involve the way in which data was collected and presented (such as MICs, evidence for OMVs, experiments with stool samples, and statistics). Most relevant, however, is the lack of convincing evidence for the role of OMVs in polymyxin resistance, such as polymyxin binding or changes in the LPS, to support the claim for OMV protection in this species.Reviewer #1:The authors explore the role of A. baumannii outer membrane vesicles (OMV) in resistance to polymyxin B (PMB). They obtain a laboratory-derived resistant strain that has altered expression of genes involved in OMV biogenesis and produces more OMVs that show protection against polymyxin B both in vitro and in vivo. This includes a MIC increase for several clinical strains and enhanced survival of A. baumannii in a Galleria mellonella assay. The results are consistent with the role of OMVs in other bacteria, contribute additional information regarding protection of A. baumannii in different experimental conditions, and suggest that OMVs protect by acting as decoys for PMB binding, although the precise mechanisms remain unclear.1. The results are interesting, but I am concerned with the specificity of the observations. In the absence of experiments with OMVs from other strains, it is difficult to assess if the results are due to this strain or if OMVs from other strains could also work (such as OMVs from the low and WT strains, the latter used also in the fecal experiment). This is also important in terms of the mechanisms involved. Given the absence of changes in surface charge (zeta potential) when cells are exposed to PMB (Figure 1C, D), what do the authors suggest is the main reason for the observed protective effect on A. baumannii cells?

We agree with reviewer’s comments. The degree of OMV production in each clinical isolate varied. Although direct correlations between the MIC and OMV production could not be identified under our tested conditions. However, higher OMV producers could show higher protection rates against PMB in the early stationary phase (Figure 6A).

The MICs of all constructed mutants were not affected, although they showed higher OMV production due to exposure to PMB at the beginning stage of culture conditions, in which no OMV production was expected. However, tendency for protecting cells grown in stationary phase from high PMC concentration (1/2 MIC of each clinical strain) was observed in our tested higher OMV producers (Figure 4F).

We performed whole genome sequencing, RNA-sequencing, proteomics, microbial community analysis, mutational analyses, MALDI-TOF, transmission electron microscopy assays, and growth tests with purified outer membrane vesicles (OMVs). In our new manuscript, we offered profound understanding that OMVs released from *Acinetobacter baumannii* functioned as decoys against PMB, which explained why other clinical isolates and microbial communities with OMV were also protected against PMB treatment. Our data suggest that OMVs can act as an important survival strategy in complex bacterial communities as well as *A. baumannii* by reducing polymyxin B binding to bacterial membrane.

2. The work with the feline microbiota is confusing and raises many questions, partly because of the complexity of dealing with mixed communities. The results indicate that certain taxa/groups (as identified by peaks) change when PMB / OMVs are administered. Some peaks seem to increase upon treatment with OMVH, while at least one peak increased when treating with OMVW, yet this is not mentioned in the text. How can this be explained? Are there no Gram-negative bacteria in this fecal sample and if so, are these affected or not by PMB/OMVs? While these experiments could suggest that these OMVs affect various cell types, couldn't addition of any antibiotic and any OMVs also trigger changes? It is therefore unclear if the changes, assumed as protection, are specific to these OMVs.

We agree with the reviewers. We previously had the T-RFLP analysis with the cat feline microbiota but recognized its inaccuracy of peak identification. Therefore, we deleted our previous T-RFLP data and performed new illumine-based bacterial community analysis using with in vitro human fecal sample.

Line 396-398: Our taxonomic profile revealed that addition of OMVs extracted from the PMR^High^ strain protected the bacterial community from PMB under our tested anaerobic conditions, resulting in recovery of a similar microbial composition (Figure 7A).

3. The authors mention binding to PMB as a strategy (eg, Lines 39-40 and Line 278), even though there is no direct evidence provided. It would be best to tone down these claims. It is also unclear how the work leads to the conclusion regarding novel antibiotics and "multifaceted" OMV production (Line 338). Please revise the sentence to more accurately reflect the findings of this study.

To address the binding issues raised in reviewer’s comments, we conducted several experiments and demonstrated direct binding of PMB to purified OMVs.

Line 358-366: To visualize direct binding PMB to OMVs, we developed a dansyl fluorophore-PMB, which was synthesized by forming a chemical bond between the primary γ-amines on the diaminobutyric-acid residues of PMB and dansyl-chloride. Confocal laser scanning microscopy (CLSM) images revealed that dansyl-PMB was less bound to PMR^High^ cells than Lab-WT cells at their respective PMB MICs (2 or 4 µg/mL), due to the fact that the PmrC-driven lipid A modification of PMR^High^ cells decreased PMB binding to OMs (Figure 5C). The lipid-selective dye (FM4-64) and dansyl-PMB were employed to visualize binding of PMB to purified OMVs of the PMR^High^ stain (Figure 5D).

Reviewer #2:In the article submitted to eLife by Shin et al., the authors investigate mechanisms of Acinetobacter baumannii resistance to polymyxin B. They isolate random mutations that occur on the genome as a result if culturing A.b. in media with PMB. They find isolates that have low and high levels of PMB resistance. Whole genome sequencing identifies nearly 20 non-synonomous mutations in different genes. Substitutions in pmrB were common between two isolates. They generate deletion mutants and overexpression genotypes for PmrB, and also contruct deletions in the other genes. The authors move on to characterize PmrB-dependent production of OMVs. The reason for focusing on this gene in favor of the others is not firmly established. The data that follow are not conducted with a sufficient level of rigor to award acceptance to this journal. No statistical analysis is presented. Significance of the data is absent, making the impact of the study very minimal.To my knowledge, the work demonstrating the involvement of PmrB in OMV formation for this important pathogen is novel and consistent with recent data that suggest modified LOS molecules might promote or prevent vesicle formation. In general the work is well written and easy to follow conceptually. However, the level of rigor applied to this study falls well short of the requirements for this journal. In its current condition, this article is not of sufficient impact to warrant publication in this journal.Data regarding PMB resistance is presented as MICs and growth curve measurements. No bacteriacidal assay is used, which is a major weakness and oversight. The authors report no cfu data for the bacteria that have been treated. The data is over interpreted.

We agreed with the reviewer. We conducted several bactericidal assays to address this issue.

Figure 4F and 6A, we conducted bactericidal assay of experimentally evolved strains, mutants and clinical isolates.

Line 387-391: After 5 days, the community cells were spotted on an LB plate and cultured under aerobic conditions to check PMB toxicity and the protective effect of OMVs (Supplementary figure 6B). The PMB-treated sample showed a 10^4^-fold reduction in aerobic bacterial cells than the control, and the OMV plus PMB sample showed similar levels of cell counts as the control (Supplementary figure 6B).

Line 518-519: The protection rate (%) was calculated survived colonies (CFU) in comparison with untreated conditions of each strain.

Measuring effects of PMB on bacterial growth on agar media is technically problematic due to the contact dependent killing activity of PMB.

We considered that the technical problem that you mentioned would not be affected to conclude our results because of several other experimental evidences which include our data with in vitro human gut microbiota. Line 351-356: The protective effect of these PMR^High^-derived OMVs was assessed using the Luria–Bertani (LB) broth (Figure 5A) and LB agar plate tests (Figure 5B), after confirming that additional purified OMVs had no effect on the bacterial growth. Interestingly, supplementation of PMB-containing liquid cultures (Figure 5A) or agar (Figure 5B) with these OMVs could increase the survival of all tested strains except for PMR^High^ on agar plates.

No statistics are provided to support significance of the findings.

Statistical analyses were added in Figure 1C, Figure 4, Supplementary Figure 1B.

The authors do not use appropriate assays to assess whether the particles they are measuring are actual OMVs. Western blots and enzymatic assays for particular proteins that localize or do not localize to the OMV are absent and must be included.

We thank the reviewer for suggesting various experimental methods. We conducted new proteomics analysis to supplement your proposal more specifically. The commonly identified OMV components were membrane-linkage proteins (OmpA, OmpW and BamE).

Line 291- 295: Gel-based proteomics with MALDI-TOF analysis were performed. Forty-three out of the 61 up- or downregulated proteins were designated with the MS-Fit program by using the NCBInr and UniPortKB databases, and 18 spots were not identified (Table S3).

Line 301-310: Interestingly, *A. baumannii* OMVs also contained many cytoplasmic cargo proteins such as ribosomal proteins and Ef-Tu, similar to the OMVs from other bacteria (Dallo et al., 2012; Deo et al., 2018). Elongation factor (Ef-Tu), observed in our proteomics analyses, is one of the most abundant proteins (>20%) in bacteria and has a canonical role in translation. However, it is also found on the surface of bacterial OMs, which suggests a possible noncanonical function on the membrane (Harvey et al., 2019; Widjaja et al., 2017). Accumulation of Ef-Tu in the periplasmic space might induce pressure, which promotes blebbing of the OM resulting in OMV secretion (Figure 3C, McBroon and Kuehn, 2007). Reduced expression of OmpA and OmpW leads to diminished linkage between the OM and peptidoglycan and subsequently causes more OMV generation in the PMR^High^

The authors offer no explanation for why the PMB resistant mutant also acquire multidrug resistance (rif, cat, amp etc.). This line of interrogation is abruptly ended. Do the other mutations contribute to this mechanism?

We stated that PMB resistant strains were not resistant to other antibiotics (rif, cat, amp etc.)

Line 165-168 : “The PMR^Low^ and PMR^High^ strains were resistant only to polymyxins, but not to the other nine tested antibiotics of different classes, including meropenem, rifampicin, chloramphenicol, and ampicillin (Supplementary figure 2).”

The model does absolutely nothing for the paper as it does not even speculate as to how a PmrB modified envelope would cause vesicle formation.

Our data suggested that either physical association of PmrB in the membrane or modification of the membrane charge by PmrB-activated PmrC might affect the production of OMVs in *A. baumannii* strains.

Lines 326-331, “when the *pmrB* gene was deleted in Lab-WT, OMV biogenesis was reduced 0.3-fold (Figures 4B–F), indicating that OMV production is impaired in the absence of PmrB. This finding was supported by our observation in which an extra copy of pmrB^H^ gene increased OMV production in Lab-WT (Figure 4B-F). PmrB/A-activated PetN transferase or incorporation of PmrB into the IM region may modulate the OM charge and integrity, inducing more OMV production.”,

The experiment where A. baumannii OMVs are added to stool samples of a cat is not rationalized sufficiently or conducted in a manner that one could specifically attribute the effects to the OMVs. A more appropriate study would involve a mouse model of disease. A. baumannii does not colonize the intestine so the relevance of this model is minimal.

We deleted our previous T-RFLP data and performed new illumine-based bacterial community analysis using with in vitro human fecal sample.

Line 394-398: Our taxonomic profile revealed that addition of OMVs extracted from the PMR^High^ strain protected the bacterial community from PMB under our tested anaerobic conditions, resulting in recovery of a similar microbial composition (Figure 7A).

Figure 7, We conducted bacterial community analysis with human fecal microbiota. Taxonomic profiling of in vitro human gut microbiomes under anaerobic conditions demonstrated that OMVs completely protected the microbial community against PMB treatment.

Fm464 staining to confirm OMV isolation is insufficient to establish that these are actual OMVs. Blotting for OmpA an OM protein and some IM protein would provide stronger support.

We developed a dansyl fluorophore-PMB and visualized direct PMB binding to purified OMVs by confocal microscopy images (Carl Zeiss, Germany). RNA-sequencing and proteomic analyses were conducted to understand contribution of genetic and proteins components (including OmpA) to OMV production. Our multi-omics studies and mutational analyses suggested that decreased membrane-linkage proteins induce overproduction of OMVs.

Similarly quantifying the lipid content with FM4-64 is not rigorous as the dye is non-specific.

We used several approaches for quantifying OMV production (lipid staining [FM4-64 dye], quantification of total proteins [Bradford assay], total DNA measurement, and LPS quantitation using the purpald assay) (Figure 4 and Supplementary figure 4B). We also developed a dansyl fluorophore-PMB and visualized direct PMB binding to purified OMVs by confocal microscopy images.

Line 637-639: The lipophilic dye FM4-64, which intercalates into the outer membrane, is commonly used to stained and observed membrane of bacteria (Rojas et al., 2018; Toyofuku et al., 2017).

Reviewer #3:This paper describes an analysis of the role of outer membrane vesicles (OMVs) in polymyxin resistance in Acinetobacter baumannii. Firstly, polymyxin-resistant A. baumannii strains were selected by growth on increasing concentrations of polymyxin and then genome sequencing showed the presence of a number of mutations. Both resistant strains had two pmrB mutations known to increase PmrC expression and give PEtn addition to LPS. The strains, especially PMRHigh, showed increased OMV production and addition of purified OMVs from this strain could increase polymyxin resistance of other sensitive strains. This is a very interesting paper worthy of further investigation. However, I feel that this paper falls short of showing a convincing role for OMVs in polymyxin resistance in this species. Without further support of presence/absence of PEtn on resistant membranes and OMVs released by the PMRHigh strain, the idea that OMVs are protective seems counterintuitive; if the LPS from this PMRHigh strain has PEtn addition (highly indicated by mutations and PmrC increased expression) then the released OMVs should not bind polymyxin due to the reduced charge. Indeed, it might be expected that OMVs from WT would be more protective as they would bind polymyxin.1. Currently, the whole genome sequencing analysis is unconvincing. Firstly, the genome sizes and number of ORFs are quite different between parent and derivative resistant strains. This seems very unusual and is likely due to different success in de novo assembly. It would have been good to use reference assembly against the parent strain as the derivatives should be almost identical. Given the likely problems with assembly, at least a number of the proposed mutations should be checked by PCR and Sanger sequencing.

Thank the reviewer for nice comments. We believed that genome sequencing is minor issue for claiming our conclusion because we performed transcriptomic and mutational analyses to understand the roles of several important genes in OMV production. The number of coding DNA sequences (CDSs) was incorrect. We corrected the number in Table S1 and sentences. The difference in the number of CDSs in resistant strains was not significant. Line 227-230: The numbers of coding DNA sequences (CDSs) in the PMR^Low^ and PMR^High^ genomes were 3625 and 3636, with mean CDS lengths of 930.2 and 947.5 respectively (Table S1).

2. The two common mutations in PmrB, and the increased expression of PmrC, indicate that polymyxin resistance is likely to be primarily due to PEtn addition to LPS and concomitant reduction in neg charge and polymyxin binding. However, this needs to be shown by LPS MS. Indeed, I think it is essential to look at the LPS structure in both the parent sensitive cells, resistant derivatives and the OMVs from each strain.

We conducted the MALDI-TOF analysis in agreement with the reviewer opinion

Figure 2, To determine whether modified lipid A plays an important role in PMB resistance, lipid A was subjected to matrix-assisted laser desorption/ionization time-of-flight (MALDI-TOF) analysis in the positive-ion mode.

Line184-190, The main peak of Lab-WT was measured at 1,490 m/z, corresponding to bis-phosphorylated penta-acylated lipid A (Figure 2A and 2B). In the PMR^High^ strain, the predominant spectrum was also measured at 1,490 m/z, followed by 1,630 m/z, which is predicted to be the peak when a PetN and a hydroxyl group were attached to the dominant lipid A (Figure 2A and 2B). The PMR^High^ strain overexpressed *pmrC* (encoding a PetN transferase), which added a PetN to some lipid A molecules, but not all, leading to interference PMB binding to the OM (see Supplementary figure 1B).

3. Mutation of pmrB in the PMRHigh strain failed to return the strain to fully sensitive, which does argue in favor of PEtn addition not being the primary reason for resistance. However, this would be a clearer experiment if the actual transferase gene (pmrC) was inactivated.

We agree with reviewer’s comments. In our new manuscript, we offered profound understanding that OMVs released from *Acinetobacter baumannii* functioned as decoys against PMB. Role of *pmrC* for PMB resistance is well-known sot that we didn’t make a corresponding mutant. Our lipid A analysis (Figure 2A), RNA-sequencing (Figure 3B), qRT-PCR data (Supplementary Figure 1B) suggested that addition of PEtN to lipid A in resistant strains, which might be responsible for PMB resistance.

4. Line 161-163. Wouldn't calculation of surface charge in the presence of polymyxin be confounded by cell death in sensitive strains?

Line 153-155: The zeta-potential analyses suggested that the average negative surface charge of PMR^Low^ and PMR^High^ decreased from – 27 mV (Lab-WT) to -20 mV and -10 mV, respectively (Figure 1B).

Line 158-161: However, this observation could not account for all the reasons underlying the high PMB resistance of PMR^High^ strain, because short-term (10 min) PMB exposure (4 μg/mL, MIC of PMR^Low^) decreased the surface charge in all tested strains, albeit to different extents (Supplementary figure 1C, n = 10 per strain).

5. Line 163-165. The fact that both resistant strains were not resistant to other antibiotics suggests in favor of PEtn addition as this is a polymyxin-specific mechanism. Why is this listed as "interesting".

Changes in surface charges on the membrane might affect the transport of other antibiotics into the membrane. “interesting” term was deleted in the manuscript.

6. What is the proposed explanation for the high number of released OMVs in the ftsL mutant but no increase in polymyxin resistance?

Figure 4, We added new protection rate analysis (tendency for protecting cells grown in stationary phase from high PMB concentration). The *ftsL* mutant was significantly protected under high PMB concentration. The MICs of all constructed mutants were not affected, although they showed higher OMV production due to exposure to PMB at the beginning stage of culture conditions, in which no OMV production was expected. However, tendency for protecting cells grown in stationary phase from high PMC concentration (1/2 MIC of each clinical strain) was observed in our tested higher OMV producers (Figure 4F).

7. Line 22 suggests that, "as expected", mutations in genes associated with cell wall cross linking such as lpp showed reduced OMV secretion. Isn't this at odds with line 106 which say that reduction in Lpp production disrupts the cross-linkage of PG on the membrane, thereby significantly contributing to the formation of OMVs.

The newly RNA-sequencing and mutational studies demonstrated that Lpp loss leads to overproduction of OMVs. We revised the sentence to clarify our intention.

Line 287-289: Notably, several genes belonging to the *pmr* and *pga* operons are included among the commonly upregulated gene categories, and the commonly downregulated genes included important membrane-associated genes (*mlaC*, *ompA*, *lpp*, and *bamE*) (Table S2).

Line 329-330: However, the *ΔftsL,Δudg, Δlpp* and *ΔmlaC* mutants exhibited higher OMV production (Figures 4A–D).

Line 334-336: Our mutational study clearly demonstrated that mutant strains lacking genes involved in membrane cross-linkages and phospholipid management (*lpp*, *ftsL* and *mlaC*) showed increased OMV secretion.

8. For conclusive assessment of the role of OMVs on altered microbiota, full metagenomic 16s rRNA should be conducted, rather than terminal restriction fragment length polymorphism analysis, which does not capture enough diversity.

We deleted our previous T-RFLP data and performed new illumine-based bacterial community analysis using with in vitro human fecal sample.

Figure 7, Analysis of fecal microbiota was conducted under aerobic/anaerobic conditions.

9. For Galleria mortality experiments, in my opinion bacterial counts should be determined

We determined the bacterial counts was 1 × 10^6^ CFU/larvae and stated that in Materials and methods section.

10. In my opinion, for conclusive understanding of the role of OMVs in decoy resistance, polymyxin binding should be assessed.

We developed a dansyl fluorophore-PMB and visualized direct PMB binding to purified OMVs by confocal microscopy images

Figure 5C-D, Dansyl-PMB were employed to visualize binding of PMB to purified OMVs of the PMR^High^ stain. Our data showed that the two dyes exhibited overlapping images, which clearly demonstrated the direct binding of PMB to OMVs.

Line 357-365: To visualize direct binding PMB to OMVs, we developed a dansyl fluorophore-PMB, which was synthesized by forming a chemical bond between the primary γ-amines on the diaminobutyric-acid residues of PMB and dansyl-chloride. Confocal laser scanning microscopy (CLSM) images revealed that dansyl-PMB was less bound to PMR^High^ cells than Lab-WT cells at their respective PMB MICs (2 or 4 µg/mL), due to the fact that the PmrC-driven lipid A modification of PMR^High^ cells decreased PMB binding to OMs (Figure 5C). The lipid-selective dye (FM4-64) and dansyl-PMB were employed to visualize binding of PMB to purified OMVs of the PMR^High^ stain (Figure 5D).

[Editors’ note: what follows is the authors’ response to the second round of review.]

Essential Revisions:The manuscript has improved with the inclusion of new data, but the authors should include more detailed information and take care with the interpretation of some of their results.1. Important discussion of other studies is missing, e.g., Jasim et al. about OMVs and polymyxins in Klebsiella (PMID: 30103446), or Macdonald et al. about OMVs in Pseudomonas (PMID: 23625841), etc.

We have added the following information to address this point:

(Lines 451-454) Notably, *Klebsiella pneumoniae* strains had outer membranes and larger OMVs harboring different lipid compositions under PMB treatment, which might contribute to the low permeability of PMB (Jasim et al., 2018).

(Lines 498-500) Two known mechanistic factors (PQS and MucD) for OMV production were not required for cycloserine-induced OMV formation in *P. aeruginosa*, indicating the existence of multiple pathways for OMV biogenesis (Macdonald et al., 2013).

Please include statistical information required to correctly interpret the results.2. Line 152: it should be noted that the observed 1.2-fold higher pmrC expression in PMRLow was not statistically significant. Indeed, 1.2-fold increase is below the usual cut-off values used for transcriptomics.

Thank you for pointing this out. The sentence has been revised to make it clear.

(Lines 160-162) Consequently, cells with more PetNs in lipid A acquire resistance to PMB. PmrC-mediated reduction of surface charges was not feasible in PMR^Low^ because the expression level of the *pmrC* gene was not significant.

3. Most of the reported transcriptomics data (e.g. Figure 3B) has no statistics associated with it, so it is not possible to know whether observed differences are significant. Add false discovery rate values or CVs to the fold-change data.

To address the statistical issues raised in the reviewer’s comments, we have added the detailed information of our RNA-seq and RPKM values in Supplementary File 1f and *Figure 3—figure supplement 1B*, respectively. Our data were calculated using log2 fold changes. In Figure 3A, new qRT-PCR analyses were conducted to verify the expression of all candidate genes (10 genes) for OMV production, which confirmed that our qRT-PCR data were consistent with the RNA-seq data.

(Line 278-281) An additional qRT-PCR assay confirmed that the *pmr* and *pgaB* genes showed more than 4-fold higher expression, but all membrane-linkage related genes (*ompA*, *bamE*, *lpp*, and *mlaC*) were downregulated in PMR^High^ (Figure 3A).

4. The number of replicates should be noted in the figure legend for all experiments. E.g. for Figure 4. it seems unlikely that there is a statistically significant difference between protein Bradford assay for LabWT and deltapmrB unless there are a large number of replicates here.

We agree with the reviewer’s comments. More than triplicate experiments were performed for all the data. We have also revised the legends of Figures and figure supplements and added statistics for all data.

5. The addition of Raman spectroscopy and the specificity of its interpretation are unconvincing.

Thank you for pointing this out. Although the Raman spectroscopy data were not convincing, this was the first time Raman spectroscopy has been used for analyzing OMVs. We removed all unnecessary information and only retained a short, important interpretation in the manuscript.

Several issues were brought up regarding the transcriptomics (in addition to statistics) and genomics data that require further clarification.6. There is no clear explanation for why different SNVs were observed in the transcriptomic data compared to the genomic data. Were these mutations found in a high percentage of the mapped reads?

To avoid any confusion, we have deleted all SNV data and removed all the related information.

7. The rationale for the RNA-Seq experiments is unclear and there were only a few genes dysregulated. Therefore, the value of this experiment is unclear.

We thank the reviewer for pointing this out. We have identified the possible candidate genes for OMV biogenesis by using RNA-seq data and confirmed the levels of their expressions by further qRT-PCR assays. Additional mutational analyses also verified their roles in OMV biogenesis. We have moved all unnecessary information and reanalyzed the RNA-seq data using a different RPKM cutoff (>50 RPKM).

8. The mutations in mRNA are troublesome as e.g., a master regulator of stringent stress response and cell division were affected and it is not clear how this is further linked to OMVs and polymyxins. Many mutations (including master regulators) in the polymyxin-resistant mutant strain could affect the observed phenotypes.

Thank you for these comments. We believe that our omics and mutational analyses substantiated that all tested genes are involved in OMV biogenesis. However, deduction of a direct link between OMV production and a single OMV-related gene might be an oversimplification because of the multifaceted control of bacterial OMV production. Further analysis for understanding how other mutations affect PMB resistance and OMV production are warranted. To avoid any confusion in the manuscript, we deleted all SNV data and removed all the related information.

9. Why do WT and PMR-resistant strains have different genome sizes? A difference of >28 kb and 11 genes sound a lot – what was deleted in these strains specifically? There are also a lot of other important mutations in the PMR high strain that could explain the growth phenotype and this makes the whole data interpretation regarding OMVs questionable.

Thank you for pointing this out. We reanalyzed our genomic data using many software programs including CLC genomics Workbench v.10.0.1 (Qiagen, Germany). We corrected all errors. The corrected genome sizes and deleted regions were marked in *Figure 3—figure supplement 1A*. Long-read NGS sequencing techniques, such as the PacBio used in this study, may produce sequencing errors. To compensate for this shortcoming, additional PCR reactions using *Pfu* DNA polymerase were performed to confirm mutational positions listed in Supplementary File 1a (18 genes). The primers used for PCR reactions were listed in Supplementary File 1k. We removed all incorrect information. Our additional transcriptomic and mutational analyses revealed that upregulation of the *pmr* operon and decreased levels of membrane-linkage proteins are linked to overproduction of OMVs, which also promoted enhanced biofilm formation. Thus, we believe that the occurrence of errors in the genome sequencing is a minor issue in relation to our conclusions.

We have also added the following discussion to address this point and new references were added.

(Line 236-239) Two chromosomal regions mostly annotated as hypothetical proteins were lost during adaptive evolution of PMR^Low^ and PMR^High^ strain (Supplementary File 1c). Linkage between those deleted regions and OMV biogenesis remains to be investigated. All 20 antibiotic resistant genes and 78 IS elements remain unchanged in both evolved strains (Supplementary File 1c, 1d).

(Line 479-487) Linkage between those deleted regions in evolved strains and OMV biogenesis remains to be investigated (Figure 3—figure supplement 1A). It is worth noting that antibiotic-induced genomic alterations including deletions, rearrangements, and amplification have been reported in many laboratory-evolved antibiotic resistant bacterial strains (Sandergren and Anderson., 2009). Several laboratory-evolved antibiotic resistant *Escherichia coli* strains also appeared to carry more than 20 mutations (Maeda et al., 2020). Tetracycline-induced genome rearrangements led to the loss of 6.1 kb having 7 full-length and 2 truncated genes in *E. coli* strains (Hoeksema et al., 2018). Norfloxacin-treated *E. coli* had ~11.5 kb deletions possibly through IS1-mediated recombination (Long et al., 2016).

10. The CRISPR-Cas9 method needs to be fully detailed in Materials and methods and it should be noted which mutants are from homologous recombination and which from CRISPR.

In lines 662-683, we have included details of the CRISPR-Cas9 method in the Materials and methods and indicated which mutants were constructed using either homologous recombination or CRISPR-Cas9.

(Lines 662-683) To construct *ΔpmrB, ΔmlaC,* and *ΔpgaB* by the CRISPR-Cas9 method, the pCasAb-apr (cat. no 121998) and pSGAb-km (cat. no 121999) plasmids were obtained from Addgene (Wang et al., 2019). A pair of 20-bp spacer oligos and 80-nt oligos were designed to target the genomic locus and donor repair template for editing. A designed 20-bp spacer was phosphorylated and annealed to the pSGAb-km plasmid with a Golden Gate assembly reaction. Then, the plasmids were transformed into *E. coli* DH5a-competent cells, followed by plating onto an LB agar plate containing 100 µg/mL, and incubated at 37 °C for 16 h. Successful cloning of the spacer was verified by PCR with the primers of Spacer-F/M13R and by sequencing with the primer of M13R (5′-CAGGAAACAGCTATGACC-3′). Then, 1M of IPTG was added into *A. baumannii* Lab-WT harboring pCasAb-apr to induce the expression of the RecAb recombination system and Cas9 nuclease. After incubating at 37 °C for 2 h, the same method was used to prepare the competent cells as described previously (Wang et al., 2019). Then, 200 ng of the spacer-introduced pSGAb-km plasmid and 300 µM ssDNA (donor repair template) were co-transformed to the IPTG-induced *A. baumannii* Lab-WT cells harboring the pCasAb-apr plasmid by electroporation. The cells were plated onto an LB agar plate containing 100 µg/mL apramycin and 50 µg/mL kanamycin, and the plate was incubated at 37 °C overnight. Successful editing was verified by PCR and sequencing. For plasmid curing, a colony was plated onto an LB agar plate containing 5% sucrose at 37 °C overnight. Then, the colony were streaked onto LB agar plates without the antibiotics to confirm curing. A detailed genome-editing protocol has been provided in the Supplementary file (*Figure 4—figure supplement 1*).

Biofilm experiments and interpretations should be further clarified. Please address the following regarding biofilm experiments:11. The biofilm data is not convincing and should be quantified for multiple areas. There is no evidence that OMVs are associated with biofilms in this study.

We would like to thank the reviewer for this important comment. To address the issues raised in the comment, we have added CLSM images (*Figure 1—figure supplement 1D*). Biofilm formation was quantified (*Figure 1—figure supplement 1E*).

(Lines 166-167) OMV-associated biofilm was measured in the Lab-WT and PMR^High^ strains (*Figure 1—figure supplement 1D and 1E*).

(Lines 170-174) Quantification of biofilm formation using the crystal violet assay also indicated that the PMR^High^ strain produced more biofilm (2-fold higher) than the Lab-WT strain (*Figure 1—figure supplement 1E*). In Figure 1—figure supplement 1G, we have included new CLSM images showing the initial stages of Lab-WT biofilm formation with OMVs.

Several published articles also have proven that OMV biogenesis is linked to enhanced biofilm formation (See our references, Baumgarten et al., 2012; Stevenson et al., 2018).

(Lines 441-443) “OMV biogenesis was thought to be also linked to the elevated hydrophobicity of the cell surface in *P. putida*, which caused greater biofilm formation (Baumgarten et al., 2012).”

(Line 264-266), “The overexpressed *pga* operon appeared to facilitate the production of PNAG-attached OMVs in *E. coli* through deacetylation of PNAG by the PgaB activity, which resulted in enhanced biofilm formation (Stevenson et al., 2018).”

In our study, enhanced biofilm formation was noticeable in all OMV overproducers and increased PgaB protein levels in the OMV were also observed in our proteomics analyses. Therefore, it is reasonable to speculate that OMV production partly contributed to the enhanced biofilm formation in our assay.

12. Supplemental Figure 1. Was the increase in biofilm seen due to excess EPS formation or an increase in viable bacteria? Was any sonication done to see if the actual number of bacteria at the timepoint chosen increase in the evolved mutant with PMB?

Biofilm formation was quantified (*Figure 1—figure supplement 1E*), and biofilm quantification (OD_680_) was normalized by the OD_600_ value, which showed that enhanced biofilm formation was not due to an increase in viable bacteria. Sonication was not used for biofilm assay.

(Lines 170-174) Quantification of biofilm formation using the crystal violet assay also indicated that the PMR^High^ strain produced more biofilm (2-fold higher) than the Lab-WT strain (*Figure 1—figure supplement 1E*).

13. There are other mechanisms of colistin (polymyxin) resistance seen including transmissible elements such as MCR-1 which encodes a phosphoethanolamine transferase leading to increased positive charge on the membrane. Is it known whether other mechanisms of resistance would also lead to increased OMV formation? If not, it might be good to mention in the discussion about whether the mechanism underlying increased OMV production would be applicable to other forms of polymyxin resistance.

Both MCR-1 gene products and PmrC function as a phosphoethanolamine (PetN) transferase, although the *mcr-1* gene is often located on the plasmid, which was not the case in our strains. Because deletion of the *pmrB* led to reduced OMV production, induction of the *pmr* operon might be linked to OMV biogenesis (Figure 4). However, there is no direct evidence for the linkage between PmrC and OMV formation. We removed unnecessary information and retained only valuable interpretative statements in the manuscript (line 324-326).

(Lines 334-336) Furthermore, when the *pmrB* gene was deleted in Lab-WT, OMV biogenesis was reduced (0.3-fold) (Figures 4A–D), indicating that OMV production is impaired in the absence of PmrB.

14. Did the authors determine whether OMVs from other pathogens would result in a similar decoy effect or would this be expected to be specific to polymyxin stimulated OMVs in Acinetobacter?

(Lines 386-390) PMB MIC tests were performed using four additional Gram-negative bacteria (*Pseudomonas aeruginosa* PAO1, *Pseudomonas putida* KT2440, *E. coli* K12, and *Acinetobacter oleivorans* DR1) and the protective roles of their OMVs were tested (*Figure 6—figure supplement 1*). Our data suggested that OMV-driven protection under PMB conditions could be applicable to other Gram-negative bacteria.

(Lines 140-143) Our multi omics data and mutational studies suggested that OMVs functioned as decoys for PMB in *A. baumannii*. Accordingly, these vesicles can protect not only the OMV producer, but also the entire bacterial community from the bactericidal effects of PMB

15. It is not clear why the authors mention DNA relaxation, but do not really explain how this is linked to whole story.

We agree with these comments. To avoid any confusion, we have deleted the sentence.

16. The document includes a large amount of experimental data that makes it hard to follow and could therefore be improved for clarity. In addition, some of the experimental information is not reflected in the abstract. Finally, authors should take care to avoid speculations.– For example, the Results section starts with a speculation about the lower growth rate, which is inappropriate but also makes the reader wonder whether other mutations could be the cause of the growth defect. It becomes clearer reading further when the WGS is described.– There is further speculation in the results that should not be there and lots of the results/discussion feel out of context.

We thank the reviewer for these encouraging comments. We have deleted and revised the unnecessary sentences and other information highlighted by the reviewer.

Please clarify some of the figures presented.17. Figure 1. Panel D TEM is not particularly quantitative. It is difficult to claim that there a larger number of OMVs in the PMR(high) strain. Was this quantitated visually in some way? If not, it might be helpful to include supplemental TEM images at a lower magnification to help support this claim.

We have included new TEM images of PMR^High^ at a lower magnification (*Figure 1—figure supplement 1F*).

18. Figure 3B (expression vs genomic position image) should be relative to log base 2 fold-change so that up- and down-regulated genes are on a similar scale.

We thank the reviewer for pointing this out. We have revised Figure 3—figure supplement 1B accordingly.

19. It would be useful to have a merged image of Figure 5D.

We have provided a merged image of Figure 5D.

20. Figure 6 Panel A. This panel is a bit confusing. I am not sure what the number scale at the top of the graph represents versus the number scale at the bottom of the panel. The dots appear blue but the authors mention black dots? I assume these are the dots they are alluding to but this needs to be clarified.

In the Figure 6 legend, “black” was rewritten as “blue.” Blue dots show the protection rate for treatment of 1/2 MIC PMB in each clinical isolate, and the bar graph shows the amount of OMVs in the early stationary phase (OD_600_ = 0.4).